# Rare mutations in the complement regulatory gene *CSMD1* are associated with male and female infertility

Arthur S. Lee [1], Jannette Rusch[1], Ana C. Lima[1], Abul Usmani [1], Ni Huang [1], Maarja Lepamets[2], Katinka A. Vigh-Conrad[3], Ronald E. Worthington[4], Reedik Mägi[2], Xiaobo Wu[5], Kenneth I. Aston[6], John P. Atkinson [5], Douglas T. Carrell[6], Rex A. Hess [7], Moira K. O'Bryan [8] & Donald F. Conrad [1,3,9]*

Infertility in men and women is a complex genetic trait with shared biological bases between the sexes. Here, we perform a series of rare variant analyses across 73,185 women and men to identify genes that contribute to primary gonadal dysfunction. We report *CSMD1*, a complement regulatory protein on chromosome 8p23, as a strong candidate locus in both sexes. We show that CSMD1 is enriched at the germ-cell/somatic-cell interface in both male and female gonads. *Csmd1*-knockout males show increased rates of infertility with significantly increased complement C3 protein deposition in the testes, accompanied by severe histological degeneration. Knockout females show significant reduction in ovarian quality and breeding success, as well as mammary branching impairment. Double knockout of *Csmd1* and *C3* causes non-additive reduction in breeding success, suggesting that *CSMD1* and the complement pathway play an important role in the normal postnatal development of the gonads in both sexes.

[1] Department of Genetics, Washington University School of Medicine, St. Louis, MO 63110, USA. [2] Estonian Genome Center, University of Tartu, 51010 Tartu, Estonia. [3] Oregon National Primate Center, Oregon Health and Science University, Beaverton, OR 97006, USA. [4] Department of Pharmaceutical Sciences, Southern Illinois University, Edwardsville, IL 62025, USA. [5] Division of Rheumatology, Department of Medicine, Washington University School of Medicine, St. Louis, MO 63110, USA. [6] Department of Surgery, University of Utah School of Medicine, Salt Lake City, UT 84132, USA. [7] College of Veterinary Medicine, University of Illinois, Urbana-Champaign, IL 61802, USA. [8] The School of Biological Sciences, Monash University, Clayton, Victoria 3800, Australia. [9] Department of Molecular and Medical Genetics, Oregon Health and Sciences University, Portland, OR 97239, USA. *email: conradon@ohsu.edu

The human genetic study of infertility has historically classified male infertility and female infertility as distinct diseases, leading to the assembly of many separate cohorts for the study of sex-specific reproductive processes[1–4]. However, numerous molecular and physiological mechanisms of fertility regulation are shared between male and female mammals[5]. For instance, programmed germ cell degeneration is a pervasive part of gonadal biology in both sexes[6,7]. In human males, roughly 80% of the meiotic descendants of spermatogonial stem cells undergo apoptosis prior to ever becoming mature spermatozoa[8]. Likewise, human females are born with ~500,000 oocytes and ovulate roughly 400 times after menarche. However, only roughly 1000 oocytes survive the sojourn to menopause, representing colossal germ cell loss not attributable to ovulation[9]. This ratio of surviving:apoptotic germ cells differs between species, but is narrowly maintained within species[8,10]. The principles that govern the loss versus survival of given germ cells are poorly understood.

Defects in germ cell development intrinsic to the gonad are defined as primary gonadal dysfunction. Primary gonadal dysfunction — which manifests as early idiopathic menopause in women and oligo-/azoo-spermia in men — is a subset of the infertility phenotype that is tractable for human genetic analysis and has a prevalence of at least 1% in adult males and females[11,12]. We previously identified a reproducible association between rare copy number variant (CNV) burden and oligo-spermia and azoo-spermia in men[13,14]. One of several candidate loci from this study included the complement regulatory CUB and Sushi multiple domains 1 (CSMD1) gene.

The primary protein sequence of CSMD1 shares homology with complement-interacting proteins[15]. Complement acts as an inflammatory/phagocytic signal in the innate immune system[16], and previous work has shown that classical complement components C1q and C3 are also responsible for microglia-mediated phagocytosis of excess neuronal cells in a normal post-natal developmental process known as synaptic pruning[17]. Common variants in CSMD1 (refs. [18,19]) and complement C4 (ref. [20]) have also been associated with schizophrenia in independent, well-powered human association studies. Furthermore, some of the most significantly associated variants previously associated with azoospermia include the greater MHC locus, which encompasses complement C2, C4, and factor B[21,22]. CSMD1 is also known to inhibit the classical complement pathway in vitro[15,23].

In this study, we incorporate array and exome sequencing data from a large cohort of early idiopathic menopause from the Women's Health Initiative[24] to search for novel, shared factors required for normal gonadal function in both sexes, identifying an association with the gene CSMD1 in male and female gonadal dysfunction, and replicate our findings with samples from the UK Biobank[25]. We characterize the expression patterns of CSMD1 in female and male gonads and measure morphological and reproductive traits in Csmd1-knockout mice, thereby adding a level of functional support to our genetic findings. Finally, to consolidate the putative roles of complement with Csmd1-mediated pathology, we investigate the in situ abundance of macrophages and complement component C3 in Csmd1-wildtype versus Csmd1-null tissues. Our findings highlight the value of human genetic analysis as an entry point into surprising and previously unappreciated pathways in disease processes — in this case, a novel connection between of complement-related processes and gonadal function.

## Results

### CSMD1 is associated with gonadal failure in both sexes. Due to the strong purifying selection against infertility mutations, we hypothesized that male and female gonadal dysfunction are driven largely by rare mutation events. To test this hypothesis, we obtained SNP array and phenotype data from 12,002 women (515 cases of early idiopathic menopause vs. 11,487 normal menopause controls) and 2,072 men (321 cases of oligo-spermia or azoo-spermia vs. 1,751 normospermic controls) with documented reproductive health history. Due to reduced linkage disequilibrium between common SNPs and rare variants, we instead leveraged the SNP log R ratios and B-allele frequencies to perform ab initio discovery of CNVs that occupy the entire allele frequency spectrum (Supplementary Data 1). We performed rigorous call and sample QC with emphasis on high confidence calls based on established methods (see Methods section[13,14,26], Supplementary Fig. 1). We then applied filters to enrich for rare, deleterious CNVs. We used these CNVs to perform a gene-based, case-control rare variant association separately in males and females. This approach allows for interrogation across the genome with a substantially lower multiple hypothesis burden than is required for traditional common-variant GWAS.

Our rare variant association study identified a significant association between early idiopathic menopause and deletions overlapping the CSMD1 gene located on chromosome 8p23.2 (OR = 16; PLINK nominal $p$-value = $4.0 \times 10^{-4}$; PLINK genome-wide $p$-value = 0.015; Fig. 1, Supplementary Tables 1, 2, Supplementary Data 2). This association signal replicated in GEMINI[13], our smaller cohort of male oligo-/azoo-spermia cases (OR = 3.3; PLINK nominal $p$-value = $6.5 \times 10^{-3}$). This CNV association is driven by an enrichment of rare deletions in cases, all of which are clustered at the 5′ end of the gene within introns 1–3 (5′-deletions; Fig. 1a).

To further replicate the association between deletions in CSMD1 and risk for gonadal dysfunction, we assembled an independent female early idiopathic menopause case-control cohort using the UK Biobank (see Methods section). After CNV QC and rigorous case/control selection, we obtained a cohort of 63,064 women with both reliable phenotype data and CNV calls (1,873 cases versus 59,947 controls). We again observed a significant association between early menopause and rare 5′-deletions of CSMD1 (OR = 3.03, logistic regression $p$-value < $5 \times 10^{-4}$, Fig. 1a, b). To summarize the overall risk conferred by rare CSMD1 5′-deletions of all sizes, we performed a meta-analysis across all three cohorts (OR = 3.6 95% CI [2.14–6.01]; meta-analysis logistic regression $p$-value = $1.2 \times 10^{-6}$; Fig. 1c; Supplementary Table 1).

We further tested our association using genotypes ascertained by an orthogonal genotyping modality, by analyzing rare (MAF < 0.01) CSMD1 single nucleotide variants (SNVs) from exome sequencing performed on a subset of the WHI cohort ($n = 1526$). We performed a single-locus quantitative trait analysis and found a statistically significant association between rare, deleterious CSMD1 SNVs and age at menopause (SKAT single-locus $p$-value < $5 \times 10^{-3}$; Fig. 1, see Methods section). The CSMD1 protein product consists almost entirely of alternating/repeating CUB (complement C1r/C1s, Uegf, Bmp1) and Sushi/CCP (complement control protein) domains (Fig. 2a). Thus, in order to assess the relative contribution of in CUB versus Sushi domain SNVs to age at menopause, we fit linear models to partition the association signal among these two domains. The CSMD1 SNV association was driven almost exclusively by mutations in the CUB ($\beta_{CUB} = -0.86$), but not Sushi ($\beta_{SUSHI} = 0.046$) domains (Wilcoxon rank-sum test $p$-value = 0.043; Fig. 1d, see Methods section). Our regression model estimates that each CUB domain SNV accelerates the onset of menopause by ~10 months. Finally, a well-powered common variant GWAS in a female cohort of 182,416 individuals identified 3 common SNPs within CSMD1 to be significantly and independently associated with age at menarche and subsequently replicated in >300,000

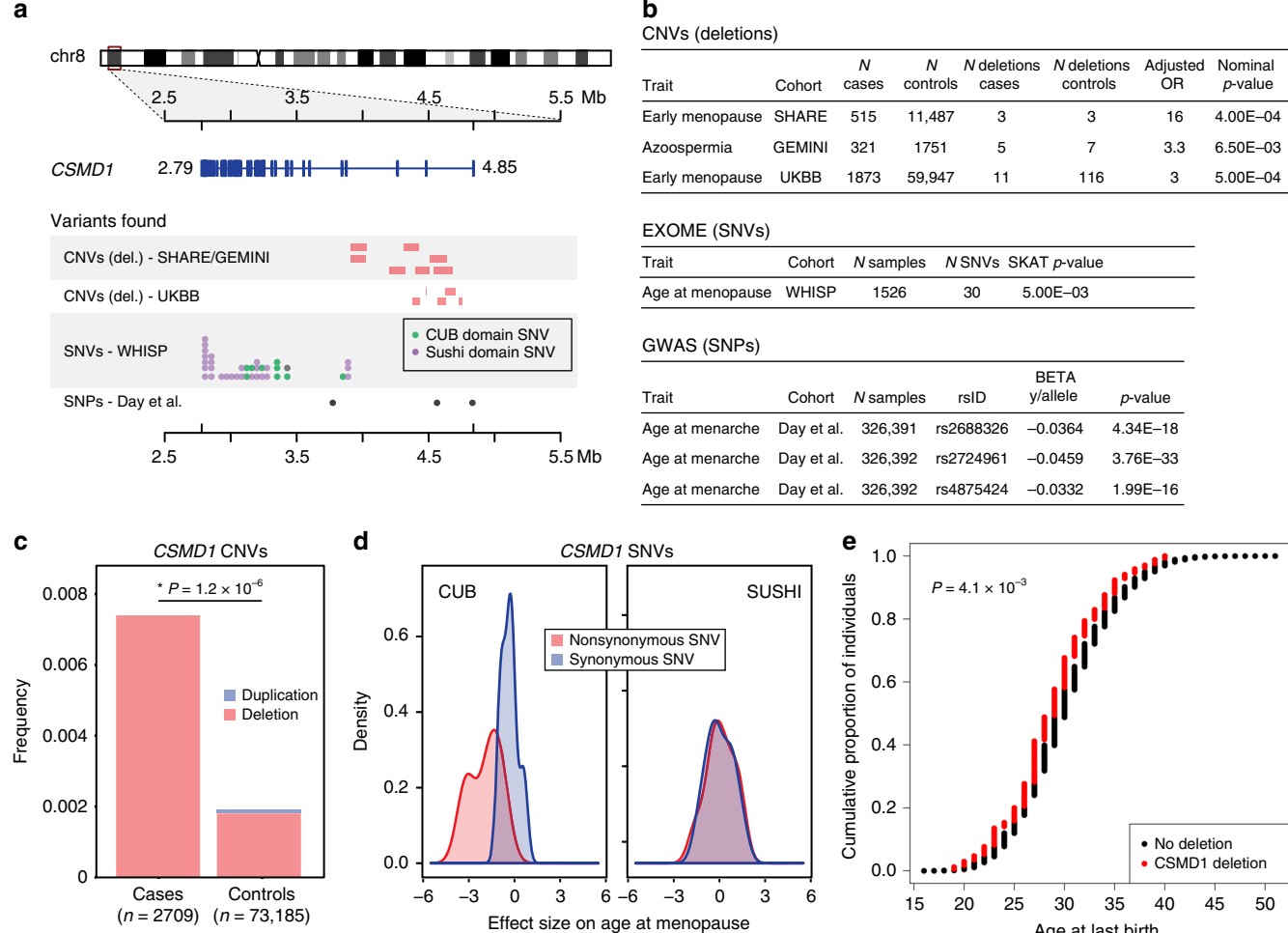

**Fig. 1** Genetic variants in *CSMD1* are associated with primary gonadal dysfunction. **a** The landscape of genetic variation across *CSMD1* among all cohorts. SHARE/GEMINI: rare deletions overlapping *CSMD1* among 14,074 females and males from the SHARE and GEMINI cohorts; UKBB: rare intron 1–3 deletions observed in 63,064 females from the UK Biobank early idiopathic menopause cohort — for clarity of visualization only the deletions observed in cases are shown; WHISP: 37 rare SNVs overlapping *CSMD1* among 1,526 exome-sequenced females; and Day et al.: three statistically independent lead SNPs from a large-scale GWAS of age at menarche in females are depicted as points along the bottom of the figure (left to right): rs2688326, rs2724961, and rs4875424[28]. All coordinates are in hg19. **b** Summary of the human genetic evidence that *CSMD1* variants are associated with gonadal function in men and women described in this study. **c** Stacked barplot depicting frequency of rare CNVs overlapping introns 1–3 of *CSMD1* among 2,709 cases of male or female gonadal dysfunction versus 73,185 controls. Rare deletions over *CSMD1* segregate significantly with cases (odds ratio = 3.6; meta-analysis logistic regression p-value = $1.2 \times 10^{-6}$). **d** Density plots depicting effect sizes (β) of rare *CSMD1* SNVs on age at menopause for synonymous (blue) and nonsynonymous (red) SNVs, stratified by protein domain (CUB vs. Sushi). SNVs in the CUB domains are significantly associated with an earlier onset of menopause when compared to SNVs in the Sushi domains ($\beta_{CUB}$ = −0.86, 95% CI [−1.56, −0.151]; $\beta_{SUSHI}$ = 0.046, 95% CI [−0.255, 0.377]; Wilcoxon rank-sum test p-value = 0.043). Synonymous mutations in both CUB and Sushi domains are centered about 0, consistent with a well-calibrated estimate of effect size. **e** The cumulative distribution of age at last birth in UK Biobank subjects, plotted separately by *CSMD1* deletion status. There is a 1-year reduction in the mean age at last birth for women with deletions in introns 1–3 of *CSMD1* compared to women without deletions (linear regression p-value = $4.1 \times 10^{-3}$). Source data are provided as a Source Data file

individuals[27,28] (Fig. 1). Subsequent work in the same cohort showed age of menarche and menopause to be positively correlated. Furthermore, the common *CSMD1* SNPs associated with age of menarche also correctly predicted age of menopause in the expected direction ($\beta_{rs2688325}$ = 0.014; $\beta_{rs7828501}$ = 0.021; $\beta_{rs7463166}$ = 0.031)[29].

The phenotype of idiopathic early menopause indicates problems in ovarian function. Predictable consequences of such ovarian dysfunction are irregular or absent menses and reduced fecundity. To further clarify the role of *CSMD1* deletions in female reproductive traits, we tested for association with 4 additional reproductive phenotypes available from the UKBB (see Methods section). Of these traits, we identified a significant association with age at last live birth (n = 76,686, linear regression

p-value = $4.1 \times 10^{-3}$, Fig. 1e). Individuals with *CSMD1* deletions show a reduced reproductive lifespan across all quartiles of the cumulative distribution, corresponding to an average reduction of 1 year. To compare relative effect sizes, our fitted model predicts that obesity status reduces reproductive lifespan by 0.6 years, and smoking status reduces reproductive lifespan by 1 year (see Methods section). The maximum age at last birth among our controls is 51 years, versus 40 years for *CSMD1* deletion carriers. However, we note that 99.6% of control individuals have completed childbearing by age 36, indicating that traditional measures of fertility (e.g. fecundity) would not necessarily capture ovarian dysfunction related to *CSMD1* deletions.

In summary, we detected associations between rare variants in *CSMD1* and gonadal dysfunction (i) across multiple classes of

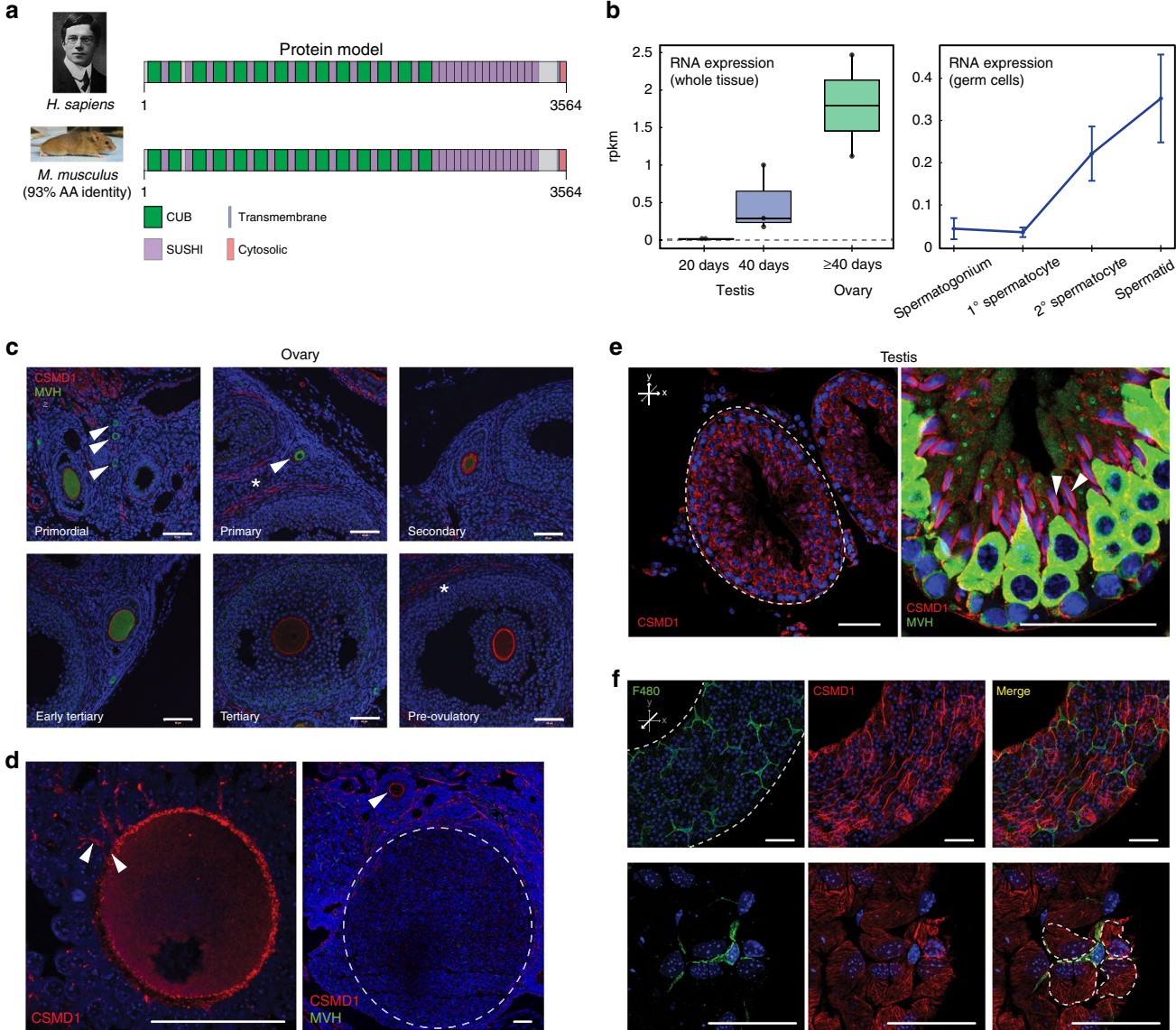

**Fig. 2** *Csmd1* is expressed in the male and female gonads. **a** Protein model of CSMD1 in human and mouse. CUB and Sushi domains, as well as the transmembrane and cytosolic domains are depicted along the protein model (97.1% of the CSMD1 protein is extracellular). **b** RNA expression of mouse *Csmd1* in sexually immature whole testes (20 days), sexually mature whole testes (40 days), and whole ovary. RNA-seq of FACS-purified germ cell populations show *Csmd1* expression changes during spermatogenesis. *Csmd1* RNA is maximally expressed at the spermatid period of development. Boxplot center line = median; hinges = upper and lower quartiles; whiskers = 1.5 × interquartile range. Line chart error bars reflect standard error of the mean. **c** CSMD1 protein immunofluorescence in developing oocytes (marked by MVH) and surrounding somatic cells. MVH expression decreases, whereas CSMD1 expression increases with follicular development. CSMD1 encircles the oocytes at all follicle stages. Follicles named in each box are marked with white arrows when necessary. Theca cells are indicated by stars. **d** CSMD1 is maximally expressed at the oocyte surface and extends into the transzonal projections (white arrowheads), which physically connect the germ cell to the surrounding somatic granulosa cells (left). During ovulation the follicle releases the oocyte and regresses to form the corpus luteum (dashed lines). CSMD1 signal is absent. An oocyte is highlighted for comparison (white arrowhead, right). **e** Immunofluorescence (IF) visualization of CSMD1 (red) in testis seminiferous tubule cross sections (x–y axis). CSMD1 protein is broadly expressed in germ cells across all stages of spermatogenesis. MVH is a primordial germ cell marker whose expression peaks early, then steadily decreases during spermatogenesis/oogenesis. CSMD1 is maximally expressed in elongating spermatids (white arrowheads). **f** Whole mount testis tubule preparation (z axis). F480-positive macrophages (green) with characteristic ramified processes occupy the longitudinal interstitial space. CSMD1 (red) is expressed in a hatched pattern which may correspond to the actin cytoskeleton of peritubular myoid cells and Sertoli cells. Four Sertoli cells surrounding a macrophage outlined by dotted lines. All scale bars = 50 μm. Source data are provided as a Source Data file

genetic variation; (ii) ascertained by orthogonal genotyping platforms; (iii) occupying multiple points along the allele frequency spectrum; and (iv) in multiple populations and cohorts.

**Detailed CSMD1 expression patterns in male and female gonads.** *CSMD1* encodes for an extremely large (>3,000 amino acid) transmembrane protein with a large extracellular portion consisting of alternating CUB and Sushi complement-interacting domains[15]. The protein encoded by *CSMD1* is conserved between human and mouse, with 93% amino acid sequence identity and 100% domain synteny (Fig. 2a). In humans, *CSMD1* is expressed in both male and female gonads (Supplementary Fig. 2), but extremely little is known of its molecular function, particularly in the context of fertility. Thus, we first performed RNA-seq on

mouse gonads and confirmed that *Csmd1* mRNA is expressed in both whole ovaries and whole testes (Fig. 2b, Supplementary Fig. 3), consistent with previous work[30,31]. Next, we detailed patterns of in situ CSMD1 protein expression through the different stages of ovulation using a validated CSMD1 antibody (Supplementary Fig. 4). Throughout oogenesis, CSMD1 shows lower expression in oocytes of early follicles (i.e. primordial and primary follicles) and higher expression in oocytes of late follicles (i.e. secondary, tertiary, and pre-ovulatory follicles; Fig. 2c). Supporting theca cells, which circumscribe the mature follicles, are also CSMD1-positive (Fig. 2c, asterisk). The post-ovulatory corpus luteum shows no specific CSMD1 expression (Fig. 2d). Female oocytes require substantial physical interaction with surrounding somatic cells[32]. At high magnification, CSMD1 is expressed along transzonal projections which connect the somatic granulosa cells to the oocyte membrane (Fig. 2d).

In the testes, *Csmd1* mRNA is minimally expressed at 20 days and more robustly expressed at 40 days of age which coincides with the onset of sexual maturity. Mammalian testes demonstrate exceptional transcriptional complexity, owing to the highly coordinated spatial and temporal synchronization required for successful spermatogenesis throughout male adult life[30]. Therefore, to capture a more detailed transcriptional portrait of *Csmd1* in testes, we purified germ cell subpopulations using established FACS protocols (see Methods section; Supplementary Fig. 3a). Subsequent RNA-seq of purified germ cells reveals low levels of *Csmd1* expression in spermatogonia and primary spermatocytes and peak expression in secondary spermatocytes and spermatids (Fig. 2b; Supplementary Fig. 3).

We also performed immunofluorescence on seminiferous tubule cross-sections which show that CSMD1 protein is expressed at the germ cell membrane across multiple aspects of spermatogenesis, including at the interface of elongated spermatids and somatic Sertoli cells, but is absent from spermatozoa, recapitulating mRNA expression patterns (Fig. 2e; Supplementary Fig. 3). While tubule cross-sections effectively capture information from the lumenal stages of spermatogenesis, much of the spatial variance of basal germ cells, Sertoli cells, and interstitial cells (e.g. macrophages) occurs along the longitudinal axis[33]. Therefore, we also performed immunofluorescence on whole-mount longitudinal preparations of individual seminiferous tubules. CSMD1 is expressed in a hatched pattern which is reminiscent of the actin bundles found at the Sertoli–Sertoli blood–testis barrier and the Sertoli-spermatid interface[34] (Fig. 2f).

***Csmd1* KO causes fertility phenotypes across multiple tissues**. Association studies in humans are effective in identifying candidates for disease risk. However, owing to the extensive linkage disequilibrium structure in the human genome, even statistically robust and reproducible associations may implicate linked but biologically non-causal loci (sometimes up to megabase scales)[35]. In order to confirm a role for *CSMD1* in mammalian reproduction, we obtained a previously described mouse strain with a knockout of the first exon of *Csmd1* (refs. [31,36]) (Supplementary Fig. 4). RNA-seq of knockout females demonstrated robust loss of *Csmd1* expression in the ovaries. In testes, we observe a more complex perturbation of expression, with absent or highly reduced expression of exons 1–57, but upregulation of exons 58–70. Finally, we cross-referenced multiple expression modalities (RNA-seq, CSMD1 immunofluorescence, β-Gal immunofluorescence, and X-gal staining; Fig. 2; Supplementary Fig. 4) and found multiple lines of evidence corroborating expected CSMD1 spatial patterns in wildtype tissues. We also identified nonspecific signal in KO tissues, in particular in our CSMD1 immunofluorescence assay (Supplementary Fig. 4c).

We generated a colony of *Csmd1* wildtype, heterozygous, and knockout mice and observed the effect of genotype on fertility, gonadal morphology and function in male and female mice. In males, *Csmd1* knockout leads to variable, but statistically significant, defects across multiple dimensions of gonad function and fertility. First, we observed an increase in infertility rate among knockout males (30%) compared to wildtype (4.5%) (linear regression $p$-value $= 4 \times 10^{-4}$, Fig. 3a). Second, knockout males show on average a 25% reduction in the daily sperm production compared to controls (Wald test $p$-value $= 0.029$, Fig. 3b). Third, a subset of *Csmd1* knockout males suffer from profound anatomic and histopathologic derangement of the testes (Fig. 3c–e; Supplementary Fig. 5). Remarkably, the most severe examples of germ cell elimination (Sertoli cell-only syndrome), could be observed as early as 34 days of age (Fig. 3c; Supplementary Fig. 5). This time point corresponds to the approximate onset of male sexual maturity and the emergence of the spermatid germ cells, where *Csmd1* is maximally expressed. Neither knockout nor wildtype testes showed evidence of derangement prior to sexual maturity (Supplementary Fig. 5c). Both histologic severity (none, mild, and severe) and age of onset (postnatal day 34 through day 300) were highly variable between individuals. In fact, different foci within the same testis of *Csmd1* knockout mice often simultaneously present different degrees of seminiferous tubule degeneration (Fig. 3d). Finally, significantly fewer germ cells express the male germ cell antigen TRA98 (Poisson regression $p$-value $< 2 \times 10^{-16}$; Supplementary Fig. 5d), in both atrophic and normal tubules, suggesting that knockout testes suffer from expression perturbations in addition to, or perhaps presaging, loss of germ cells and frank degeneration. Together, these observations indicate that the *Csmd1* knockout mutation is not fully penetrant and may be influenced by environmental and/or stochastic events. However, even after accounting for age covariates, testes derangement status segregates significantly with *Csmd1* genotype (multivariate ANOVA $p$-value $= 7.69 \times 10^{-3}$; see Methods section; Fig. 3e). Lastly, we performed serial backcrossing for 9 generations on a subset of mice to validate the effect of the *Csmd1* null allele on an approximately constant genetic background (see Methods section). We recapitulated the degeneration phenotype in these backcrossed male knockouts (Supplementary Fig. 5e), indicating that *Csmd1* genotype status — not genetic background — drives this signal of degeneration.

Detailed histological analysis of 50 knockout testes revealed heterogenous and distinct classes of degeneration in *Csmd1*$^{-/-}$ testes (Fig. 3d). Spermatogenesis begins to become disorganized, especially at the late steps of spermiogenesis, with failure of spermiation, fewer numbers of elongating spermatids in the lumen, and ectopic mixing of spermatid steps in stages IX–XII. This is followed by the sloughing of all types of germ cells into the lumen; remaining germ cells can be observed in abnormal tubules that appear to be missing one or more waves of spermatogenesis, and these eventually resolve as Sertoli cell-only tubules. Sloughed germ cells can be seen downstream in the epididymis, and, occasionally they obstruct the downstream rete testis leading to dilation of the upstream tubules.

Three non-exclusive scenarios may account for the net loss of germ cells in *Csmd1* knockouts: (i) increased germ cell death; (ii) failure of germ cell proliferation/maturation; and (iii) increased phagocytic clearance of germ cells. We did not observe obvious differences in the abundance of apoptotic germ cells apparent by H&E staining. We excluded systemic endocrine defects that would be observed in the case of failure of the hypothalamus or pituitary (Supplementary Fig. 6). We also did not observe any developmental stage-specific accumulation or depletion of germ cells as characterized in maturation arrest/failure of meiosis[37,38].

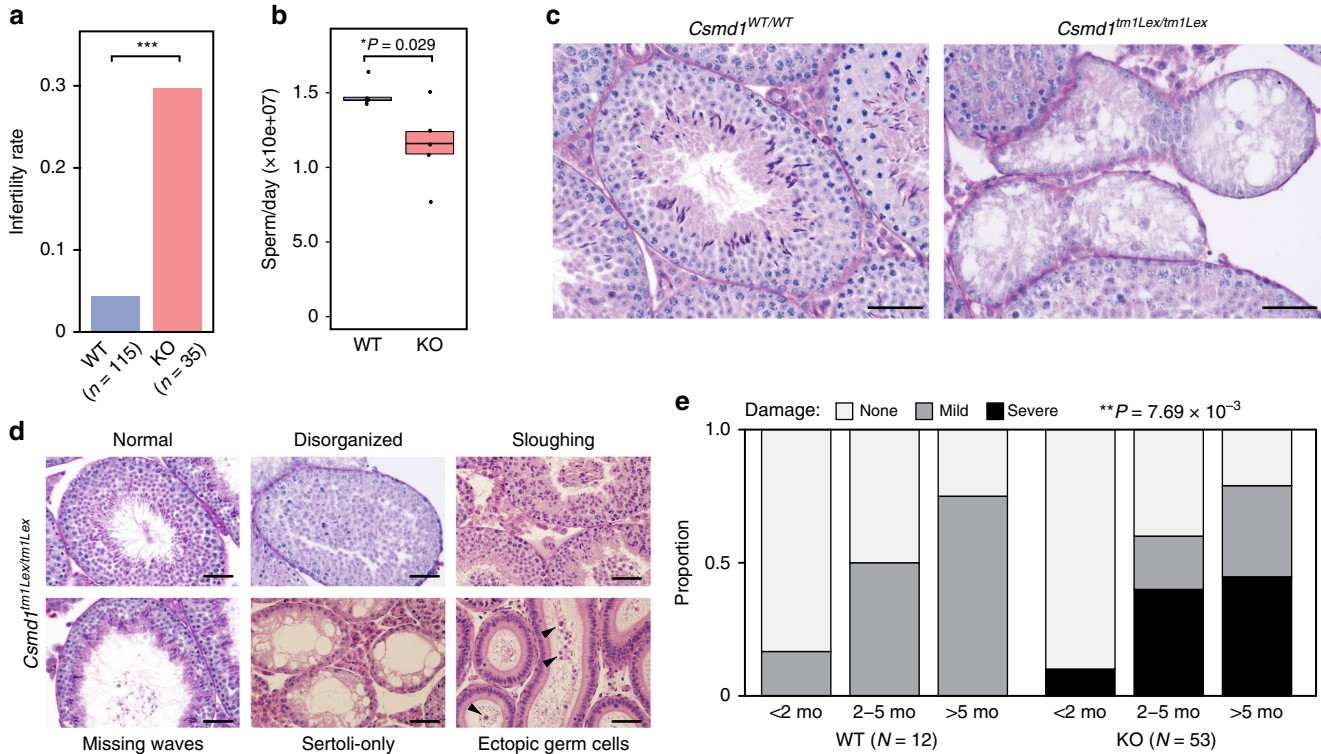

**Fig. 3** Variable reproductive defects in *Csmd1* KO males. **a** Barplot depicting proportions of infertile *Csmd1* wildtype (WT) vs. knockout (KO) males. KO males showed significantly higher rate of infertility (relative risk = 6.8, linear regression *p*-value = $4 \times 10^{-4}$). **b** Daily sperm production is 25% lower in knockout testes after controlling for age (Wald test *p*-value = 0.029). Center line = median; hinges = upper and lower quartiles. **c** Seminiferous tubule histology of wildtype vs. knockout littermates at postnatal day 34. The majority of knockout tubules in this sample contain no germ cells (Sertoli-only). **d** Qualitative classes of progressive morphologic degeneration. Seminiferous tubules from *Csmd1* knockout males showing normal morphology (Normal); loss of spatiotemporal structure, but retained germ cells in all phases of spermatogenesis (Disorganized); loss of early germ cells into the tubule lumen (Sloughing); loss of all germ cells except late-step spermatids (Missing waves); and loss of all germ cells, leaving a signature of vacuolization (Sertoli-only). Sloughed germ cells from upstream tubules are occasionally captured in the downstream epididymis (Ectopic germ cells). *Csmd1* knockout males can display multiple classes of degeneration even within the same testis. Like the histological phenotypes, fertility status varies greatly among knockout individuals, with some males becoming infertile within weeks, and others within months. **e** Quantification of the degeneration phenotype. Testis sections from *Csmd1* wildtype and knockout animals were assigned one of three possible histological degeneration scores: None = 0, Mild = 1, or Severe = 2 (see Methods section). The stacked barplots depict the proportion of damaged tubules among wildtypes and knockouts, stratified by age group. All proportions sum to 1. Damage severity segregates significantly with genotype after controlling for age (ANOVA *p*-value = $7.69 \times 10^{-3}$). All scale bars = 50 μm. Source data are provided as a Source Data file

Finally, we did not observe significant differences in the proliferation marker PCNA levels between *Csmd1* wildtype (9.4%) and knockout (7.0%) testes.

**Complement C3 deposition is increased on *Csmd1* KO testes.** FACS-purification and RNA-seq of wildtype male germ cells show that *C3* and *Csmd1* mRNA expression are negatively correlated throughout spermatogenesis (Fig. 4a); in contrast to *Csmd1*, *C3* mRNA is most highly expressed in spermatogonia and decreases precipitously in subsequent stages. In wildtype testes, we observed abundant signal of both C3 protein and macrophage marker F4/80 in the interstitial space between seminiferous tubules, but never across the blood–testis barrier (Fig. 4b). Next, we estimated macrophage abundance and C3 protein deposition in *Csmd1* wildtype vs. knockout testes (Fig. 4c; Supplementary Fig. 7). The percentage of C3-positive deposition is significantly higher in *Csmd1* knockout testes than in wildtype ($\bar{x}_{wt} = 1.7\%$; $\bar{x}_{ko} = 6.6\%$; ANOVA *p*-value = $7.7 \times 10^{-4}$). C3 is a secreted, circulating protein whose mRNA is expressed by multiple cell types — including the germ cells themselves. Therefore, we also measured *C3* mRNA expression in *Csmd1* wildtype versus knockout purified male germ cells (Fig. 4d). *C3* mRNA expression does not differ between wildtype and knockout germ cells of any type, indicating that the

germ cells are not necessarily the source of the increased C3 protein deposition.

**_Csmd1_ KO ovaries are reduced in quality.** The observed rate of infertility in knockout females (3/37, 8.1%) was not significantly higher compared to controls (5/115, 4.3%, Fig. 5a). The female infertility phenotype we ascertained in human does not likely represent congenital infertility, but instead adult-onset early menopause, which is difficult to model in mice[39]. Thus, we searched for more subtle gonadal defects associated with *Csmd1* genotype. We estimated female time to pregnancy based on retrospective husbandry records. From these records, we generated a well-calibrated null distribution of time to conception in females of all genotypes which demonstrates distinct periodicity lasting 4–5 days, recapitulating the periodicity of the mouse female estrous cycle (Fig. 5b). Next, we stratified our population by maternal genotype. For wildtype mothers, the bulk of conceptions occurred within the first estrous cycle as expected[40], whereas most *Csmd1* knockout mothers required a time equivalent to two or more cycles to achieve pregnancy ($\beta_{GT} = 10.4$; linear regression *p*-value = 0.012). A small minority of knockout females required many cycles to achieve pregnancy (>60 days). Circulating gonadotropin levels did not differ between wildtype and

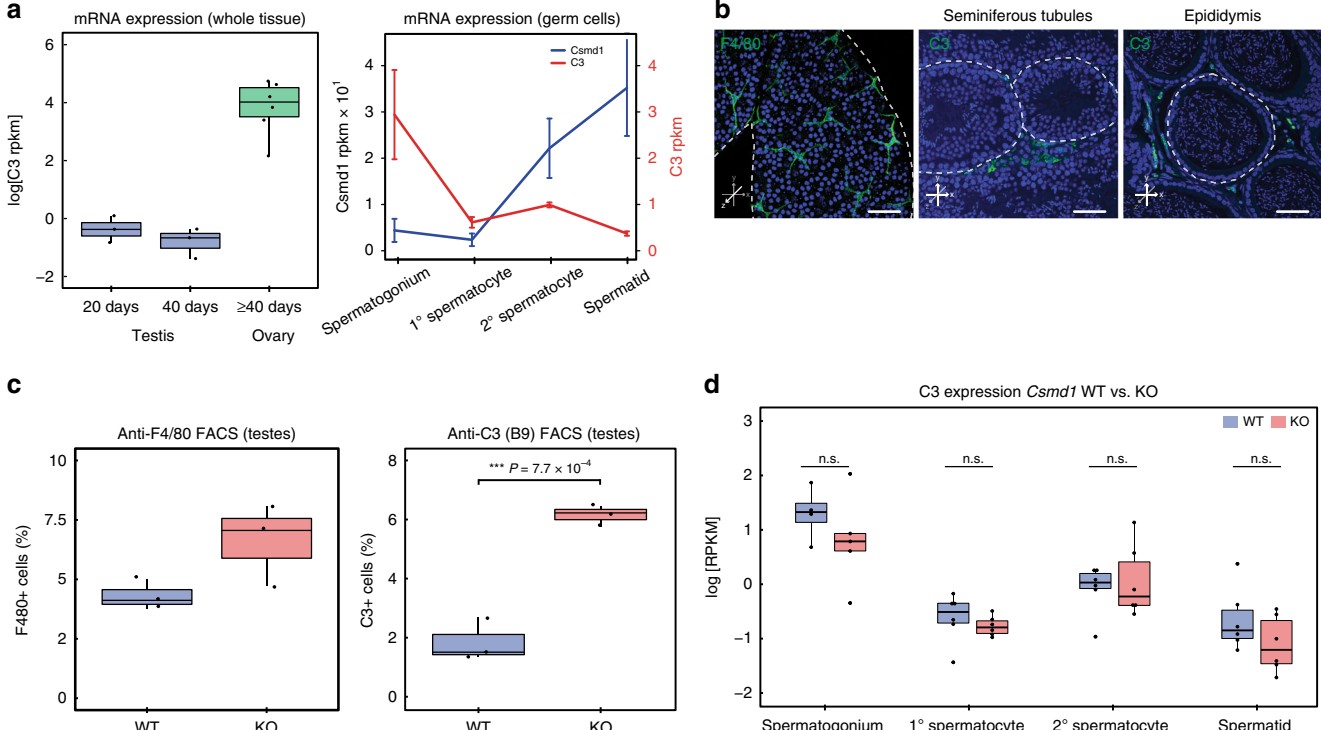

**Fig. 4** Normal and aberrant complement C3 localization in *Csmd1* wildtype versus knockout testes. **a** *C3* mRNA expression in sexually immature whole testes, sexually mature whole testes, and whole ovary. RNA-seq of FACS-purified germ cell populations shows *C3* expression decreases throughout spermatogenesis. *Csmd1* expression data from Fig. 2 are rescaled and superimposed for ease of comparison. Line chart error bars reflect standard error of the mean. **b** Complement and macrophages are confined to the interstitial compartment in normal tubules. F4/80 immunofluorescence of whole mount tubule preparations shows abundance of macrophages along the interstitium (left). C3 immunofluorescence of tubule cross-sections shows expression in the interstitium, but not within the lumenal compartment (middle). Cross-section of downstream epididymis shows continued exclusion of C3 from the lumenal compartment (right). Individual tubules are circumscribed by dashed lines. Scale bars = 50 μm. **c** Boxplots depicting F4/80 abundance and C3 protein deposition in FACS-sorted *Csmd1* wildtype vs. knockout testes. Both F4/80 and C3 are increased in *Csmd1* knockout testes, though only significant for C3 (ANOVA *p*-value = $7.7 \times 10^{-4}$). **d** RNA-seq of *Csmd1* wildtype vs. knockout germ cell populations shows no significant difference in *C3* mRNA expression (two-tailed *t*-test; *p*-value > 0.05). For all boxplots, center line = median; hinges = upper and lower quartiles; whiskers = 1.5 × interquartile range. Source data are provided as a Source Data file

knockouts after controlling for estrous stage, suggesting that this reduction in mating success was not secondary to impaired hormonal input (Supplementary Fig. 6). *Csmd1* knockout females had significantly smaller ovaries by mass when controlling for age, (quadratic regression *p*-value = $8.1 \times 10^{-3}$; Fig. 5c; Table 1), and knockout females showed significantly more atretic follicles and fewer normal pre-ovulatory follicles at necropsy (Hotelling *t*-test *p*-value = $3.5 \times 10^{-3}$; Fig. 5d). In a subset of female knockout ovaries, we observed inflammatory changes associated with infiltration of lipid-engorged macrophages (foam cells) and occasional ovarian cysts (Fig. 5e, f). Altogether, we find that *Csmd1* knockout females display ovarian defects that are normally associated with progressive wildtype aging (i.e. smaller ovary size[41], increase in atretic follicles[42], and enrichment of foam cell macrophages[43]), but at significantly accelerated rates and higher proportions in knockouts.

**C3 is abundant in follicles and colocalizes with CSMD1.** In wildtype ovaries, C3 is localized to the oocyte surface in normal developing follicles, colocalized with CSMD1, and then observed to be diffused in large amounts throughout the corpus luteum (Fig. 6), which does not express CSMD1. Macrophages are a prominent cell type in the ovarian stroma but are excluded from early stage follicles; they later invade corpora lutea and degrading follicles[44]. Furthermore, C3 colocalizes with macrophages in the

corpus luteum, atretic follicles, and among developing follicles to variable degrees. C3 is also abundant within the early follicular antrum, suggesting that C3 may be important for remodeling the connections between granulosa cells during antrum formation (Fig. 6b). Previous work has shown that activated C3 is present in human antral follicular fluid at levels comparable to sera[45].

**Csmd1 KO mothers have impaired mammary branching.** Interestingly, while knockout females achieved fewer pregnancies per estrous cycle, the average number of offspring born per successful pregnancy did not differ significantly between wildtype and knockout mothers ($\bar{x}_{wt} = 6.6$ (95% CI [5.4−7.8]); $\bar{x}_{ko} = 6.9$ (95% CI [5.7−8.1]); Table 1). However, pups borne of *Csmd1* knockout mothers suffered from significantly higher mortality rates during the neonatal period (1–10 days) when compared to wildtype/heterozygous mothers (% mortality$_{WT+het}$ = 10.5% (95% CI [3.6−17.5%]); % mortality$_{KO}$ = 50.0% (95% CI [30.0−70.0%]); Poisson regression *p*-value = $7.93 \times 10^{-7}$; Fig. 7a). We performed necropsy on expired offspring which revealed an absence of milk spots, suggesting death by starvation. Because neonatal mortality segregated with maternal genotype but not offspring genotype or paternal genotype, we hypothesized that this increase in mortality was caused by nursing defects in *Csmd1*-deficient mothers. CSMD1 is expressed in the normal mammary gland through the adult life cycle of wildtype females

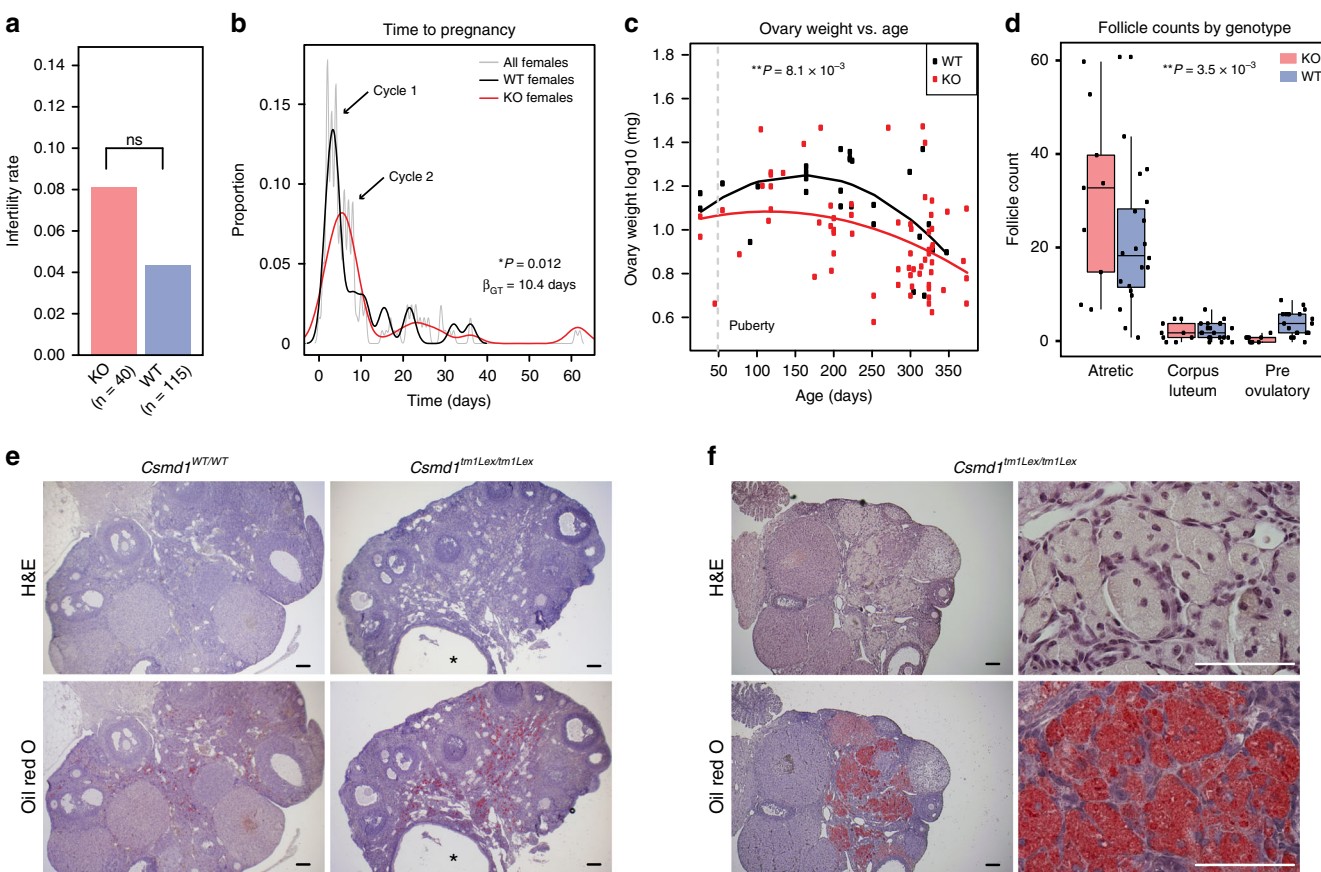

**Fig. 5** *Csmd1* knockout ovaries show reduced morphologic quality and reproductive performance. **a** Barplot depicting proportions of infertile *Csmd1* wildtype (WT) vs. knockout (KO) females. In the population sampled ($n = 115$ WT + 40 KO), *Csmd1* genotype does not significantly predict infertility status in females (relative risk = 1.9, linear regression $p$-value > 0.05). **b** Probability density plot depicting female mating success over time, stratified by maternal genotype (black = wildtype females; red = knockout females; gray = all females). The distribution is periodic, recapitulating the periodicity of the mouse female estrous cycle. Knockout females require significantly more cycles to achieve pregnancy ($\beta_{GT} = 10.4$, linear regression $p$-value = 0.01), with a small subset of females never achieving pregnancy. All statistical models account for confounders such as age, body weight, and male factors when appropriate (see Methods section). **c** Scatterplot depicting ovary weight versus age, stratified by genotype (black = wildtype females; red = knockout females). The fitted polynomial models are depicted as lines with color corresponding to genotype. Knockout females have significantly smaller ovaries than wildtype when controlling for age (quadratic regression $p$-value = $8.1 \times 10^{-3}$). Gray hashed line demarcates rough onset of puberty in females. **d** Composition of follicles differs significantly between wildtype and knockout female ovaries, with significantly fewer pre-ovulatory follicles and significantly more atretic follicles in knockout (Hotelling $t$-test $p$-value = $3.5 \times 10^{-3}$). Center line = median; hinges = upper and lower quartiles; whiskers = 1.5 × interquartile range. **e** Ovarian histology in a wildtype versus knockout female. Knockout ovaries were consistently enriched for foam cell macrophages, confirmed by Oil Red O stain in adjacent sections. We occasionally observed ovarian cysts in knockout females (asterisk). **f** Ovary from 336-day-old knockout showing extensive involvement of foam cells occupying 40% of the tissue (left). High magnification of ovary section from same animal; top shows multinucleated appearance of foam cells, bottom is Oil Red O stain of adjacent section (right). All scale bars = 100 μm. Source data are provided as a Source Data file

## Table 1 Raw biometry and fecundity measurements from *Csmd1* mutant colony

|  | WT | KO |
| --- | --- | --- |
| Ovary weight | 0.057 [0.024−0.090] | 0.028 [0.017−0.039] |
| Mammary weight | 0.48 [0.35−0.61] | 0.37 [0.26−0.48] |
| Testis weight | 0.29 [0.24−0.34] | 0.27 [0.25−0.29] |
| Maternal litter size | 6.6 [5.4−7.8] | 6.9 [5.7−8.1] |
| Paternal litter size | 5.4 [3.4−7.4] | 7.4 [6.6−8.2] |

Mean values +/− 1.96 standard errors of the mean. All weights are the sum of bilateral measurements per animal in grams. Litter size measurements are stratified by maternal and paternal genotype and do not include individuals that failed to conceive (i.e. they only include non-zero litter sizes)

on both lumenal epithelial cells and myoepithelial cells of the mammary ducts, and on numerous stromal cells (Fig. 7b, c). Mammary glands from knockout females showed reduced density of the epithelial branching network during mid-pregnancy and post-nursing (Fig. 7d). Quantification of duct morphology in nulliparous wild type and knockout animals shows a significant reduction of lateral branching prior to pregnancy (Fig. 7e, f). Finally, we measured C3 and CSMD1 expression patterns in wildtype mammary glands (Fig. 7g). As early as puberty, C3 can be seen in high levels within the mammary duct lumen of virgin animals. C3 is also expressed within vesicles of specific subsets of CSMD1-positive stromal cells (Fig. 7g).

***Csmd1*−/− *C3*−/− double KO males and females are infertile.** Based on previous findings that CSMD1 is a negative regulator of C3, we predicted that removal of C3 would partially or completely

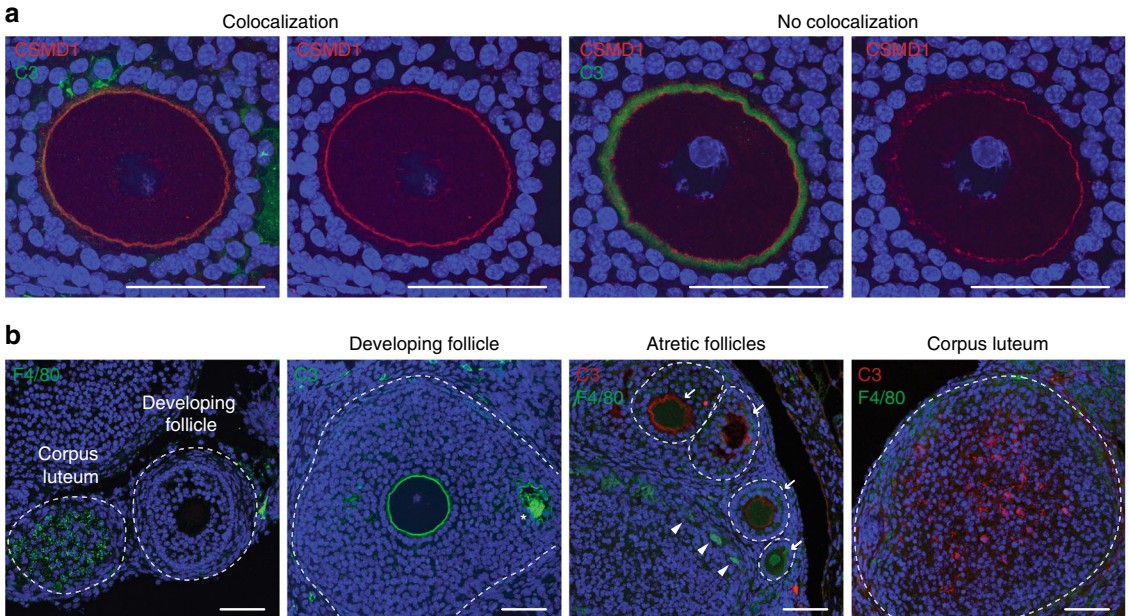

**Fig. 6** C3 and CSMD1 protein colocalize at the oocyte surface. **a** Spatial relationships between C3 and CSMD1 in developing follicles. C3 and CSMD1 immunofluorescence of oocytes in developing follicles. In most follicles, C3 and CSMD1 colocalize at the oocyte plasma membrane with overlapping signal (Colocalization, left panels). Occasionally the two signals separate and C3 is localized in a tight but distinct ring outside of CSMD1 (no colocalization, right panels). **b** Spatial relationships between C3 and F4/80-positive macrophages in developing, atretic, and post-ovulatory follicles. F4/80 immunofluorescence of adjacent follicles shows positive macrophage signal in corpus luteum, but not within developing follicles. C3 is localized to the oocyte membrane, as well as in the follicular fluid of the developing antrum (asterisk). Atretic follicles at different stages of degeneration (white arrows) show varying levels of C3 and F4/80 expression. F4/80 can also be seen in a punctuated pattern along the stroma and thecal layers (white arrowheads). C3 is also present in corpus luteum. Individual follicles are circumscribed by dashed lines. All scale bars = 50 μm. Source data are provided as a Source Data file

alleviate the morphological degeneration and fertility defects observed in *Csmd1* knockout mice. To test this prediction, we generated colonies of $C3^{-/-}$ single KO mice and $C3^{-/-}$ $Csmd1^{-/-}$ double KO mice. We did not observe any severely deranged $C3^{-/-}$ testes or ovaries (Supplementary Fig. 8). Surprisingly, we found no evidence of rescue of the $Csmd1^{-/-}$ single KO phenotype in double KO males or females (Supplementary Fig. 9). Instead, we observed an unmasked phenotype of more severe histological degeneration in all DKO females, characterized by far more extensive invasion of foam cell macrophages, extensive pyknosis, and deformed follicles. We also observed loss of cellularity in the mucosal layer of the oviduct (Supplementary Fig. 9b). We monitored the fertility of 19 DKOs (10 males and 9 females) and, of these, only 4 (21%) produced progeny after at least 3–7 months of mating (3 males and 1 female; Supplementary Fig. 9, Supplementary Table 3). These extreme phenotypes are not observed in *Csmd1* or *C3* single KOs[46], suggesting that the combined effect of *Csmd1* and *C3* on fertility is synergistic.

## Discussion

We used a rare variant association approach to search for genes that modulate both male and female gonadal function, and identified the complement regulator *CSMD1* to be reproducibly associated across multiple patient cohorts and different classes of genetic variation. Our study was powered to capture rare variants of mild to moderate effect — that is, mutations that significantly increase risk of developing infertility, but still potentially segregating in the population. Conversely, our study was poorly powered to identify extremely rare/de novo, highly penetrant infertility mutations (e.g. *TEX11* (ref. [38]), *SYCE1* (refs. [38,47]), *SPIDR*[48]). We found that multiple types of genetic perturbations

of *CSMD1* associate with infertility traits in humans, most notably deletions within introns 1–3 of the gene. Intronic deletions can affect gene function by altering splicing, RNA stability, and transcriptional regulation, and recent work has shown that such deletions tend to reduce transcript levels when they perturb expression of their host gene, although upregulation is also possible[49]. All infertility-associated deletions found in this study fall within 500 kb of 3 independent GWAS peaks for age at menarche. In the GTEx atlas, we found that only two tissues show significant *CSMD1* expression-associated quantitative trait loci (eQTL): thyroid and testis[50] (Supplementary Fig. 10). Although the *CSMD1* gene body spans over 2 Mb of the genome, all of these eQTL are clustered within a 390 kb window spanning introns 1 and 2. The 3 noncoding GWAS SNPs from Day et al.[28] are not in appreciable LD with the GTEx eQTL. We also screened ENCODE[51] for functional annotations in this region. Functional summaries (e.g. chromHMM output) are not yet available for testis, ovary or mammary gland; instead we used functional summaries derived from 6 human cell lines and identified 4 consensus CTCF sites within the region (Supplementary Fig. 10). We conclude that *CSMD1* introns 1 and 2 likely harbor functional elements that can influence gene expression in testis, but additional functional data are needed to map these elements, as well as potential elements in ovary and mammary gland.

Furthermore, we found that rare protein-coding SNVs within the CUB domains of CSMD1 are significantly associated with age at menopause. According to the Residual Variation Intolerance Score (RVIS) only 0.169% of genes in the genome are more intolerant to protein coding changes than *CSMD1* (ref. [52]). Likewise, *CSMD1* is massively depleted for rare loss-of-function (LoF) variation in the gnomAD reference database[53], with a pLI score of 1.0. This depletion of LoF variation is especially striking

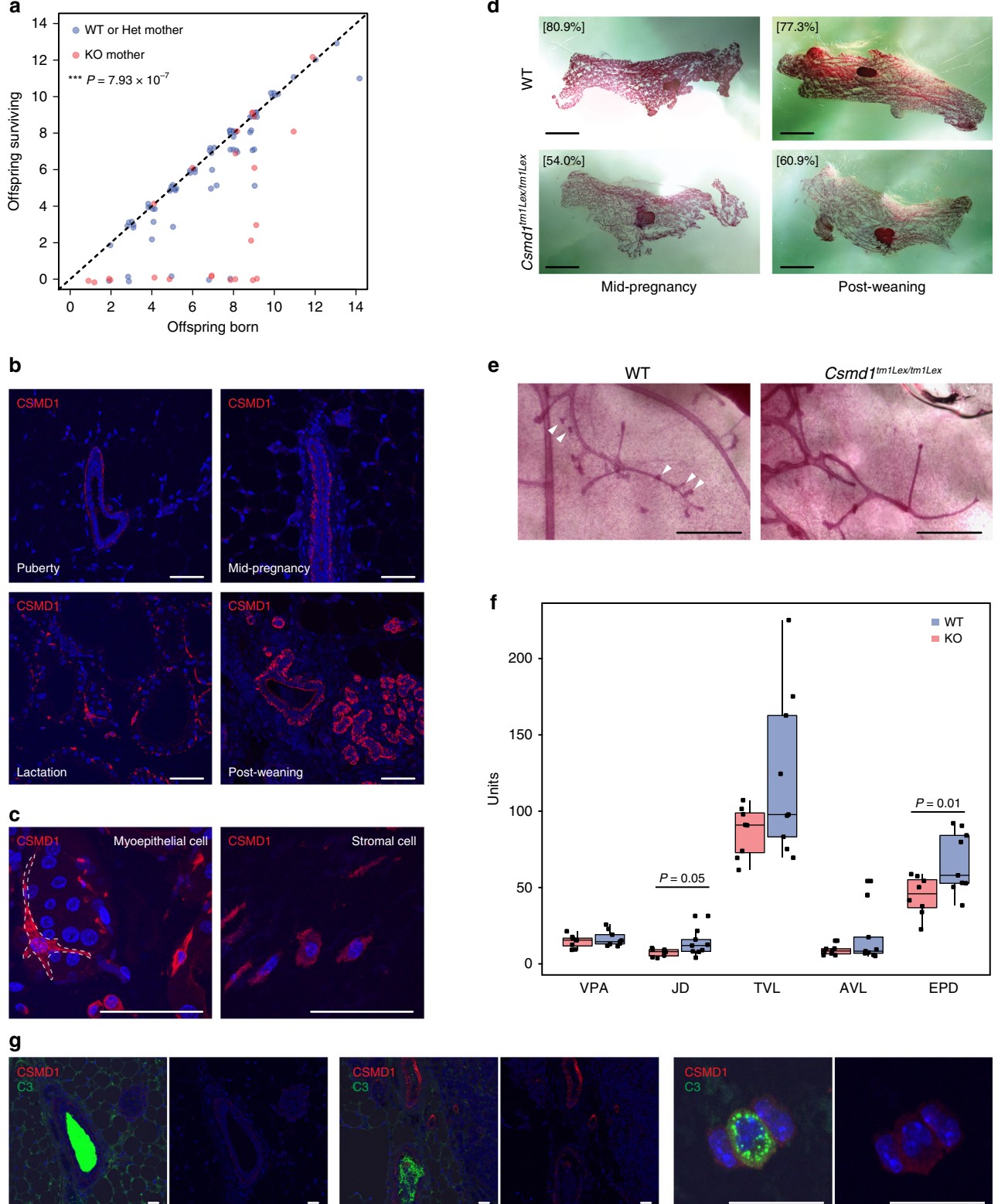

given the local increase in *CSMD1* de novo SNV mutation rate[54,55], which is also corroborated by the substantial enrichment of rare synonymous variation in *CSMD1* (ref. [53]). Altogether we favor a LoF effect of *CSMD1* mutations in human primary gonadal dysfunction.

How do our mouse studies relate to the human genetic findings and what are their limitations? We characterized a *Csmd1* genetic knockout construct that was not strictly identical to any of the

rare patient mutations found in our study. Further complicating our interpretation of the mouse data is our finding that *Csmd1* KO gonads display nonspecific and/or residual expression signal; a phenomenon previously observed in *Csmd1* KO brain[36]. Given the exceptional intolerance of *CSMD1* to LoF variation in humans, we hypothesize that incomplete *Csmd1* loss-of-function is buffered by functional redundancy[56] (e.g. from *Csmd2* and *Csmd3*)[57], and that complete loss-of-function of *Csmd1* would

**Fig. 7** *Csmd1* KO mothers have increased offspring mortality related to maternal nursing defects. **a** Scatterplot depicting number of pups surviving the neonatal period versus live births, stratified by maternal *Csmd1* genotype. Points that lie along the dashed line (slope = 1) represent litters with no neonatal deaths. Maternal *Csmd1* genotype is significantly associated with surviving litter size (10.5% vs. 50.0% survival; Poisson regression *p*-value = $7.93 \times 10^{-7}$). Points deviate slightly from whole numbers for ease of visualization. **b** Immunofluorescence shows CSMD1 expression in bifurcating mammary ducts and bulbous terminal end buds. CSMD1 is expressed on both lumenal epithelial cells and myoepithelial cells throughout different stages of adult life. Expression is lowest at puberty and increases during pregnancy, with the highest intensity during involution. Scale bars = 50 μm. **c** Higher magnification image of CSMD1 expression in a myoepithelial cell (dashed lines, left) surrounding an alveolus during lactation and stromal cells (right). Scale bars = 50 μm. **d** Whole-mount mammary glands of wildtype and knockout littermates show reduced epithelial network density (square brackets) in knockouts. Scale bars = 5 mm. **e** Knockout ducts demonstrate reduced number lateral branches (arrowheads). Scale bars = 500 μm. **f** Image analysis in age-matched wildtype vs. knockout nongravid nulliparous adult mammary ducts. Knockout females have significantly reduced number of branch points along the ducts (JD, two-tailed *t*-test p-value < 0.05) and reduced number of end segments (EPD, two-tailed *t*-test p-value < 0.01). VPA = percent area occupied by ducts. JD = branchpoint density per mm. TVL = sum of Euclidean distances among adjacent branchpoints. AVL = average Euclidean distance between adjacent branch points. EPD = number of duct end points normalized by total vessel length. Center line = median; hinges = upper and lower quartiles; whiskers = 1.5 × interquartile range. **g** C3 and CSMD1 colocalize in mammary. Immunofluorescence shows abundant C3 signal within mammary ducts with empty lumina (left) and ducts with cells in the lumen (middle). C3 signal is also present in some CSMD1-positive stromal cells (right). Scale bars = 20 μm. Source data are provided as a Source Data file

lead to much more severe and/or systemic phenotypes in both humans and mice. A complete tissue-specific and/or inducible loss-of-function knockout of *Csmd1* could help to better disentangle these potential effects but would be technically prohibitive due to the size of the gene.

Despite these practical limitations, the mouse mutants provided an invaluable and previously undescribed functional link between *CSMD1* and reproductive function. Our human samples were ascertained based on pathologic loss of germ cells, and/or reduced reproductive lifespan. We observed heterogeneous gonadal defects corresponding to reduced reproductive lifespan and premature germ cell loss in both male and female *Csmd1* KO mice, and these phenotypes were unexpectedly exacerbated in *Csmd1*$^{-/-}$, *C3*$^{-/-}$ double KO mice. While we have neither explicitly nor implicitly defined the mechanism of *CSMD1* mutations on gonadal function here, we have constructed a framework that implicates overzealous activation of the complement pathway as a potential contributor to the pathophysiology of the disease, due to the known molecular function of CSMD1 as a complement inhibitor[15], the spatiotemporal patterns of complement-related proteins and cell types in the gonads, and the excess complement deposition on *Csmd1* KO testes.

While complement has a long-appreciated role in innate immunity, it also plays a prominent role in the regulation of self cells during normal postnatal development. A classic example of this phenomenon is complement-mediated synaptic pruning in the developing brain, a process that is under genetic influence and can confer risk for disease when dysregulated[20,58]. Here, we have reported abnormal post-natal development in *Csmd1* knockout animals for three additional tissues; we have observed complement and CSMD1 protein expression in all three tissues, and previous work has shown that macrophages are essential for proper development in all three tissues. A parsimonious model to describe the set of defects we observe here is that macrophages (and potentially other phagocytes such as Sertoli cells) regulate and refine developing cells in testis, ovaries, and mammary by controlled deposition of complement onto their cell surface, accompanied by subsequent phagocytosis.

Specifically, in the testis, increased C3 deposition in *Csmd1* knockout taken in context of the known complement-inhibitory function of CSMD1 (ref. [23]) suggests that overzealous complement-mediated phagocytosis of developing germ cells by macrophages and/or Sertoli cells underlies at least part of the testicular defects. C3 mRNA and protein are much more highly expressed in the interstitial spermatogonial layer, which is physically sequestered from the lumenal aspect of the tubules by the blood–testis barrier;[59] Conversely, CSMD1 is largely expressed in

the adlumenal compartment (Fig. 2e, f). Spermatogonial fate is closely tied to interstitial macrophages[33], which are abundant in the testis, express factors essential for spermatogonial differentiation, and enter the tubules and phagocytose germ cells during aging and disease[60,61]. Concerted apoptosis of differentiating spermatogonial stem cells is a prominent and quantitatively reproducible feature of normal spermatogenesis[62]. If complement marks some targets of phagocytosis in the testis in an analogous manner to that shown in the brain[63], ectopic complement activation across the blood–testis barrier may inappropriately activate the apoptotic and phagocytic apparatus in *Csmd1* KO testes. Remarkably, *TEP1* (a distant ortholog of *C3*) has been shown to clear apoptotic germ cells in the mosquito testis by this very process[64].

Macrophage activity in the ovary is very carefully regulated in time and space during the estrous cycle. Prior to ovulation, macrophages are excluded from the granulosa layer of follicles of all types — except the atretic follicles, which are invaded by macrophages and do not ovulate[65–67]. After ovulation, macrophages invade the ruptured follicle which undergoes apoptosis/phagocytic luteolysis, forming the corpus luteum[68]. We showed that CSMD1 is also highly expressed on oocytes of the developing follicle, but not in the corpus luteum (Fig. 2d). *CSMD1* loss of function may allow for premature macrophage invasion of the developing follicle, leading to excessive oocyte atresia, fewer ovulations, and reduced probability of pregnancy (Fig. 5).

In addition to the gonads, *CSMD1* may also participate in post-natal developmental processes across other tissue systems. We have demonstrated a robust association between neonatal mortality rate and maternal *Csmd1* genotype status, with corresponding reduction in the epithelial network of the maternal mammary gland (Fig. 7). The mammary gland is a highly motile network of branching epithelial tissue that advances and recedes during different stages of post-natal development (i.e. puberty, pregnancy, nursing, etc[69].). The characteristic directionality of mammary branching is conferred by polarized cell proliferation and phagocytosis mediated by macrophage remodeling, especially in anticipation of nursing[70]. Furthermore, multiple complement and complement-regulatory components are robustly upregulated during periods of apoptosis and phagocytosis in the mammary tissue of multiple species including humans[71,72], though the functional significance of this regulatory program is unknown. Breast milk itself also suppresses complement activation[73]. Finally, CSMD1 is expressed on the lumenal aspect of mammary ducts and terminal end buds, where much of the pregnancy-associated breast remodeling occurs[15,74] (Fig. 7b). We observed a reduction in mammary epithelial density, due to reduction in

secondary or tertiary branch points, whose normal geneses are governed by multiple upregulatory and downregulatory chemotactic signals in concert with physical interaction with phagocytic cells (i.e. macrophages)[75].

The unmasking of a more severe phenotype in *C3/Csmd1* DKO mice is an unexpected but previously documented signature of complement-mediated disease. For example, double knockout of complement factor H (*CFH*) and factor P (*CFP*) unexpectedly converts mild C3 glomerulonephritis to lethal C3 glomerulonephritis in mice[76]. Similarly, *CFH/C3* DKO unexpectedly unmasks a more severe form of age-related macular degeneration in mice[77]. Multiple explanations for this phenomenon have been set forth, including a dual role of C3, differences between fluid-phase and local C3 activation, and context-specific C3 gain-of-function. Finally, we cannot exclude the possibility of primary defects outside of the gonads causing pathology in our double knockouts (e.g. altered hemostasis, central defects, etc.). More extensive mutation constructs including conditional knockouts and allelic series may help to distinguish among these scenarios.

Altogether, our human genetic and model organism findings lend support to a role of *CSMD1* across multiple tissue types, and unveil a potential role for the complement system in postnatal developmental processes across multiple tissues in the body. When interpreted in context of existing observations from mammalian brain and other model organisms, we predict that macrophage mediated complement activity on self cells is a normal and highly controlled process in many developmental systems in metazoans. Furthermore, a wealth of literature shows that genetic dysregulation of complement and complement-regulating factors is a core feature of several heritable diseases beyond infertility;[20,58,78–87]; we therefore consider *CSMD1* to also be a reasonable candidate for many of these diseases. Our work highlights the need for deeper investigation into the role of immune system components in reproductive tissues, and the opportunities that such work can have to illuminate and connect common biological processes that produce disease in more complex contexts across the body.

## Methods

**Human patient populations.** All human genetic studies were reviewed and approved by the Institutional Review Board of Washington University in St. Louis, under protocol numbers #201107177 and #201109261 and all participants provided informed consent. Male idiopathic infertility cases were clinically ascertained and pre-screened for chromosome Y deletions and compared against normospermic controls[13,14]. WHI-SHARe. To create an analogous case-control cohort of female gonadal dysfunction, we turned to the SNP Health Association Resource (SHARe)[88]. We constructed early idiopathic case and control definitions from the dense reproductive phenotype data collected on each subject. A self-reported age of menopause before 40 years was used as the only case inclusion criterion. Case exclusion criteria were oophorectomy prior to age 40, a diagnosis of lupus or rheumatoid disease, and a "yes" answer to the question "Did a doctor ever say that you had cancer, a malignant growth, or tumor?". Smoking history, which is a known factor influencing ovarian reserve, was controlled for during the analysis of genetic data.

UK Biobank. This research has been conducted using the UK Biobank Resource under Application Number '17085'. We generated a table of phenotype data for constructing early idiopathic menopause case and control labels using controlled-access data from the UK Biobank. Exclusion criteria for the study were: withdrawn consent, poor heterozygosity or missingness as defined by the UK Biobank; >10 relatives in the UK Biobank cohort; not used in autosome phasing, apparent sex chromosome aneuploidy; mismatch between genetic and self-reported sex; ever smoker; any self-reported non-Caucasian ancestry; oophorectomy prior to age 40; prior diagnosis of rheumatoid arthritis, lupus, or pelvic cancer; mismatch between self-reported ethnicity or age at menopause among three assessments; SNP array call rate <98%. In the case of pairs of 2nd degree relatives or closer, the one individual with the lower SNP-call rate was dropped. The inclusion criteria for POI case status were self-reported age of menopause <40 years old, and all remaining individuals in the cohort (after exclusions mentioned above) were used as controls.

**Mouse colony breeding.** We acquired a constitutive *Csmd1* knockout mouse (*Mus musculus*) on a mixed 129SvEvBrd:C57BL/6 background from the UC Davis

KOMP Repository (Project ID CSD118901)[89]. A 1.086 kb deletion encompassing *Csmd1* exon 1 and part of intron 1 were replaced with a lacZ/neomycin cassette. Deletion of this segment was confirmed with Southern blot and PCR. Due to the extreme size of *Csmd1* (1.6 Mb), we also analyzed RNA seq data across all 70 exons in knockout testes and ovaries. In ovaries, knockout read counts relative to wild-type are suppressed across all 70 exons. In testes, knockout read counts relative to wildtype are broadly suppressed across exons 1-57 and upregulated from exons 58-70. The amino acid coding portion of these upregulated exons range in size from 45 bp to 180 bp. The translational viability of these fragments is unknown. All littermate tissue comparisons in this study (described below) were generated from $dam_{heterozygous} \times sire_{heterozygous}$ crossings from this original colony. Next, to eliminate variance in phenotype explained by variance in background genotype (if any), we serially backcrossed the *Csmd1* mutation onto a constant C57BL/6 background for 5 generations. From this F5 backcross generation we performed a $dam_{heterozygous} \times sire_{heterozygous}$ cross from this to create wildtype and knockout littermates, and performed analogous histology and immunofluorescence experiments as with the original colony (described below; Supplementary Fig. 5e). We performed micro-satellite genotyping of these littermates to estimate the C57BL/6 background after backcrossing (Washington University Rheumatic Disease Core). We estimated the F5 proportion of C57BL/6 ancestry of 0.91 (95% CI [0.89−0.93]). For DKO experiments, we introgressed a *C3* mutant line[90] until we achieved *Csmd1/C3* DKO mice. All animal experiments were performed in compliance with the regulations of the Animal Studies Committee at Washington University in St. Louis under protocols #20120244 and #20160089.

**CNV and SNV discovery.** Array data for the Women's Health Initiative SHARe cohort were downloaded from the NCBI Database of Genotypes and Phenotypes (dbGAP accession number phg0000g1.v2). SHARe samples were genotyped on the Affymetrix 6.0 platform. We created a high-quality set of CNV calls for all cohorts using our own internal pipelines. SHARe samples were processed with Affy6CNV (a wrapper that we wrote for the Birdsuite package[91]) for data processing and QC. We obtained raw SNP array data from the UK Biobank and performed single sample CNV discovery using PennCNV[92]. Individuals with >200 CNV calls were dropped. CNV calls with PennCNV quality score >30 were retained, and adjacent CNVs in the same sample were merged. All qPCR validation of CSMD1 CNV calls were performed on Applied Biosystems TaqMan assay # Hs0367742_cn (catalog #4400291). We used Human RNase P for the reference assay (catalog #4403326). We used standard Applied Biosystems TaqMan protocols in 10 μL reaction volumes. We generated 4 to 8 technical replicates for each sample in a 384 well plate. PCR conditions were as follows: 50 °C for 2 min, 95 °C for 10 min, then 40 cycles of 95 °C for 15 s and 60 °C for 1 min. TaqMan copy number was estimated using the CopyCaller software and compared to PennCNV copy number estimates.

Exome sequencing was performed on a subset of the WHI subjects as part of the Women's Health Initiative Sequencing Project (WHISP); all available WHISP BAM files were downloaded from the NCBI Database of Genotypes and Phenotypes (dbGaP accession phs000200.v10.p3.c1 and phs000200.v10.p3.c2)[93]. Genotypes were recalled, jointly, from 1668 WHI BAM files using Haplotype Caller, recalibrated and cleaned according to GATK best practices using GATK-3.2.2.

**Association testing.** CNVs. Rare CNVs were associated with case-control status using generalized linear models. For the SHARe association analysis, we included the top 10 ancestry eigenvectors (calculated from the full SHARe genotype matrix) and smoking history as covariates. For the UK BioBank association analyses, we included as covariates BMI, smoking history (or individuals with any history of smoking were excluded, where noted) and the top 10 ancestry eigenvectors calculated from the full UK Biobank SNP genotype matrix.

In addition to idiopathic early menopause (defined above in Human patient populations), we performed association testing for the following 4 traits for UKBB subjects (Unique Data Identifier codes in parentheses): Length of menstrual cycle (3710), Number of live births (2734), Age at last live birth (2764), all-cause female infertility IC10 code (N97).

SNVs. We tested for an association between rare SNVs in *CSMD1* and age at menopause in the WHI samples using the Sequence Kernel Association Test (SKAT)[94], weighting each variant with the Combined Annotation Dependent Depletion (CADD) value[95]. Five ancestry eigenvectors and smoking history were included as covariates; significance was evaluated using bootstrapping with 5000 samples.

**Testes dissociation and cell sorting and RNA extraction.** We adapted existing methods for dissociation and germ cell purification[96] which we describe below. Sexually mature (40 ± 1 days old), male mice were sacrificed, and their testes were decapsulated and homogenized in a 1× MEM solution (Gibco 11430-030) containing 120 U/mL Type I Collagenase (Worthington Biochemical LS004194) and 1 mg/mL DNAse I (Roche 10104159001), and agitated for 15 min. 1× MEM was replaced and added with 50 mg/mL Trypsin (Worthington Biochemical 54J15037) and 1 mg/mL DNAse I and agitated for 15 minutes, then mechanically homogenized for 3 min. 50 mg/mL Trypsin and 1 mg/mL DNAse I were added and agitated again for 15 min. We added 0.4 mL heat inactivated Fetal Bovine Serum

(Sigma F1051), 5 μL Hoescht 33342 (Life Technologies H3570), and 1 mg/mL DNAse I, and agitated for 15 minutes. Individual cells were dissociated by pipetting sequentially through two 40 μm cell strainers (Falcon 352340). For each individual mouse, one dissociated testis was used for whole tissue RNA extraction and sequencing, and the other testis was used for germ cell purification, RNA extraction, and sequencing. All dissociation steps were performed at 33 °C. Dissociated testes were sorted on a modified MoFlo cytometer (Beckman Coulter) at the Washington University Siteman Flow Cytometry Core using a krypton-ion laser[97]. Cells that are stained with Hoechst can be clustered in two wavelengths: (i) blue fluorescence, which is informative of DNA content, and (ii) red fluorescence, which is informative about chromatin state and Hoechst efflux from the cell. Our gating strategy is graphically summarized in Supplementary Fig. 3. Based on these parameters, we separated homogenized testes suspensions into four purified populations: (i) spermatogonia, (ii) primary spermatocytes, (iii) secondary spermatocytes, and (iv) spermatids. These separated populations were collected and RNA extraction performed on them. RNA from whole testes was extracted with the RNeasy Plus Mini Kit (Qiagen 74134), and RNA from FACS-purified germ cell populations was extracted with the RNeasy Plus Micro Kit (Qiagen 74034).

**RNA-seq.** Whole testis, whole ovaries, and purified male germ cell subpopulations were obtained from wildtype and *Csmd1* null siblings (summarized in Supplementary Data 3). We extracted polyadenylated mRNAs from each tissue/cell type and converted these into RNA-seq libraries. Three biological replicates of each tissue or cell type were sequenced with a 2 × 101 bp paired-end protocol. Reads were mapped to Ensembl *Mus musculus* reference R72 and transcript expression levels were summarized as reads-per-kb of exon per million-mapped reads (RPKM) using the TopHat2 package[98]. RPKMs were adjusted for batch effects and cryptic covariates using PEER[99], quantile normalized, and then the R package poissonSeq was used for differential expression analyses[100].

**Immunostaining and imaging.** Testes and ovaries were dissected, fixed in 4% paraformaldehyde (Electron Microscopy Sciences), and embedded in paraffin. We baked 5 μm sections at 60 °C for 1 h, deparaffinized in Xylenes, and rehydrated into PBS (Corning). Antigen retrieval was done in boiling citrate buffer (10 mM sodium citrate, 0.05% Tween-20, pH 6.0) for 20 min. Sections were blocked in PBS containing 0.2% Triton X-100 and 5% normal donkey serum (Jackson Laboratories) for 1 h at room temperature and then incubated with primary antibodies diluted in blocking solution over night at 4 °C. After washing with PBS-Tx (PBS containing 0.2% Triton X-100), they were incubated with fluorescent secondary antibodies in blocking solution for 1 h at room temperature, washed with PBS-Tx, and treated with 0.2% Sudan Black in 70% EtOH for 10 min. The sections were then counterstained with Hoechst dye 33342 diluted 1:500 in PBS for 5 min, washed once with PBS-Tx for 2 min and then with PBS, and mounted in ProLong Diamond anti-fade mounting medium (Molecular Probes). Imaging was done on an Olympus LSM700 confocal microscope using Zen software, and images were processed using Photoshop CS5 (Adobe). Antibodies used were gt α-CSMD1 N20 (Santa Cruz Biotechnology sc-68280 1:100), rb α-mouse vasa homolog (MVH) (Abcam 13840, 1:1,000), donkey α-gt CF594 (Biotium #20116 1:300), and donkey α-rb Alexa488 (Life Technologies #A-21206 1:300), rb α-β-gal (Cappel #55976 1:333), rat F4/80 BM8 (Santa Cruz Biotechnology sc-52664 1:50), donkey α-rat Alexa488 (Life Technologies #A-21208 1:300), rb α-C3 (Abcam 200999, 1:2000), and gt α-rb Alexa568 (Life Technologies #A-11011, 1:300). Testes were decapsulated in 1x PBS and the tubules were gently teased apart and fixed in 4% paraformaldehyde in 1× PBS for 2 h at 4 °C. After washing with PBS and blocking in PBS containing 0.2% Triton X-100 and 5% normal donkey serum for 1.5 h at room temperature, they were incubated in primary antibodies overnight at 4 °C in blocking solution and processed as the cross-sectional IF experiments were described above[33]. For immunohistochemistry, 5 μm paraffin sections were treated as above, except the secondary antibody was biotin-coupled horse α-goat (Vector Laboratories, BA-9500, 1:200), and detection was done using the Vectastain Elite ABC kit (Vector Laboratories, PK-6100) and DAB Peroxidase Substrate kit (Vector Laboratories SK-4100) per the manufacturer's instructions. Sections were counterstained with hematoxylin, mounted in Cytoseal Xyl (Thermo Scientific), and imaged on a Zeiss Axioplan 2 microscope equipped with an Olympus DP71 camera and DP software. For X-gal staining, whole tissues were fixed in 4% PFA in PBS for 2 h, washed with PBS, and stained in X-gal (GoldBio #X4281C1) overnight[101]. Tissues were embedded in paraffin, sectioned at 5 μm, and counterstained with hematoxylin.

**Histology.** Freshly-dissected gonads were fixed under agitation in Modified Davidson's fixative (Electron Microscopy Sciences 64133-50) for 24 h and Bouin's fixative (Electron Microscopy Sciences 26367-01) for 24 h. Fixed tissues were embedded in paraffin and sectioned at 5 μm. Sectioned tissues were stained with hematoxylin and counter-stained with either Periodic acid-Schiff reagent[102] or eosin. Stained testes from 65 individual mice of known age and genotype (12 wildtype, 53 knockout) were provided to a single mouse pathologist in a blinded fashion. All samples received a score of 0 (no damage), 1 (mild damage), or 2 (severe damage) (see Supplementary Fig. 5a for exemplars). In order to estimate the

effect of genotype on score, we fit a linear analysis of variance model:

$$y_{ijk} = \mu + \alpha_i + \beta_j + \varepsilon_{ijk} \qquad (1)$$

where $y_{ijk}$ is the damage score for individual $k$, $\mu$ average damage score across all animals, $\alpha_i$ is the effect of genotype $i$, $\beta_j$ is the effect of age $j$, and $\varepsilon_{ijk}$ is the random error associated with the $k$th observation.

**Germ cell quantification.** We performed immunofluorescence as described above on a pair of 34 day old male littermates (the same individuals as seen in Supplementary Fig. 5c) using TRA98 antibody (Abcam ab82527 1:200). We generated count data for total cells (filtering based on size and shape), and for TRA98-positive cells (filtering based on green fluorescence) using the ImageJ software package. In order to estimate the effect of genotype on TRA98 cell count, we fit the following model:

$$\ln(y_i) = \beta_0 + \beta_1 X_{1i} + \beta_2 X_{2i} + \varepsilon_i \qquad (2)$$

Where $y_i$ is the TRA98-positive count in image $i$, and $X_1$ is the genotype (*Csmd1* wildtype versus knockout), and $X_2$ is the total cell count. $\varepsilon_i$ is the nuisance variable for image $i$.

**Gonad size analysis.** We sacrificed 229 adult mice (106 males and 123 females), and measured body weights and bilateral gonad weights at necropsy. For males, mean body weight was 37.1 g, mean testes weight was 273 mg, and mean age was 201 days. For females, mean body weight was 31.3 g, mean ovary weight was 32 mg, and mean age was 234 days. In order to estimate the effect of genotype on gonad weight, we fit the following linear model:

$$y_i = \beta_0 + \beta_1 X_{1i} + \beta_2 X_{2i} + \beta_3 X_{3i} + \varepsilon_i \qquad (3)$$

Where $y_i$ is the gonad weight in individual $i$, and $X_{1i}$, $X_{2i}$, and $X_{3i}$ are the genotype, age, and body weight of individual $i$, respectively. $\varepsilon_i$ is the nuisance variable for individual $i$.

**Quantifying daily sperm production.** Daily sperm production (DSP) was estimated using the Triton-X 100 nuclear solubilization method[103]. Frozen testes were quickly thawed at room temperature and approximately half was used to estimate DSP. Sample weight was registered for full testis and tissue fragment used for the assay. To burst all cells, except for spermatid heads, the tissue was sonicated in DSP buffer (0.9% NaCl, 0.01% sodium azide (SAZ), 0.05% Trition-X-100) using short pulses (3 × 10 s at 30% amplitude). Sperm heads were then collected by centrifugation (10 min at 500×g) and used in 1:1 dilutions with 0.4% Trypan Blue for estimation of cell density per homogenate in a hemacytometer. These values were then used to estimate the total number of spermatids per testis [number spermatids in homogenate divided by sample weight (g) and multiplied by the full testis weight (g)]. In mouse spermiogenesis, steps 14–16 take 4.84 days (developing spermatids), therefore, DSP was calculated by dividing the total number of spermatids in one testis per 4.84 days.

**Follicle count analysis.** We sacrificed 15 sexually mature female mice, of which 10 were wildtype and 5 were knockout genotypes. Bilateral ovaries were fixed, sectioned to 5 μm, and stained with H&E. We identified primordial follicles, primary follicles, secondary follicles, early antral follicles, antral follicles, preovulatory follicles, atretic follicles, and *corpora lutea* across each section based on histologic appearance[104]. In order to estimate the effect of genotype on gonad weight, we fit the following model:

$$\ln(y_i) = \beta_0 + \beta_1 X_{1i} + \beta_2 X_{2i} + \varepsilon_i \qquad (4)$$

Where $y_i$ is the number of total oocytes in bilateral ovaries of individual $i$, and $X_{1i}$ and $X_{2i}$ are genotype and age, respectively. $\varepsilon_i$ is the nuisance variable for individual $i$.

**Breeding time analysis.** We compiled comprehensive husbandry information over a period of greater than 1 year corresponding to 151 litters born representing all possible *Csmd1* wildtype, heterozygote, and knockout sire/dam breeding combinations. We calculated the number of days between first sire/dam co-habitation and birth of each litter. Next we subtracted an estimated C57BL/6 gestation time of 19 days[105] to estimate time to conception. We also calculated parental ages at conception. All density plots depicted in Fig. 5b reflect estimated time to conception for all 151 litters. In order to estimate the effect of maternal genotype on mating success, we controlled for paternal genotype by including wildtype sires only. We then fit the following linear model:

$$y_i = \beta_0 + \beta_1 X_{1i} + \beta_2 X_{2i} + \beta_3 X_{3i} + \varepsilon_i \qquad (5)$$

Where $y_i$ is the estimated time to conception for mating pair $i$, $X_{1i}$ is maternal genotype (wildtype, heterozygote, or knockout), $X_{2i}$ is maternal age at conception, and $X_{3i}$ is paternal age at conception. $\varepsilon_i$ is the nuisance variable.

**Litter size analysis.** We bred 44 females (8 wildtype, 27 heterozygote, and 9 homozygote) with 41 males (4 wildtype, 26 heterozygote, and 11 homozygote) over

a period of 10 months to produce 99 litters, totaling 688 live births. All 9 parental genotype permutations [$wt_{dam} \times wt_{sire}$, $wt_{dam} \times het_{sire}$ … $hom_{dam} \times hom_{sire}$] were represented multiple times (excepting $het_{dam} \times wt_{sire}$). We counted deaths in during the neonatal period (defined as 1–10 days by convention, although the vast majority of deaths occurred within 24–48 h) and subtracted from the live birth total to obtain the final number of surviving pups (550 total). Next, we stratified each litter by maternal and paternal genotype status (Csmd1 wildtype or heterozygous versus knockout) and fit the following model:

$$\ln(y_i) = \beta_0 + \beta_1 X_{1i} + \beta_2 X_{2i} + \varepsilon_i \qquad (6)$$

Where $y_i$ is the number of surviving pups in litter $i$, and $X_{1i}$ and $X_{2i}$ are the maternal and paternal genotypes, respectively. $\varepsilon_i$ is the nuisance variable for litter $i$.

**Mammary gland whole-mount analysis.** Female littermates were collected at four developmental time points: (i) pre-pubescent (<30 days of age); (ii) adult virgins; (iii) mid-pregnancy (estimated 14 days after copulation); (iv) post-weaning (7 days after weaning pups from mother's nursing). Freshly-dissected whole inguinal mammary glands were fixed overnight in Carnoy's solution (60% ethanol, 30% chloroform, 10% glacial acetic acid). Fixed tissues were washed and rehydrated in ethanol and water and stained in Carmine alum histological stain (0.5% Aluminum potassium sulfate, 0.2% Carmine) for 48 h. Stained tissues were dehydrated with increasing concentrations of ethanol and stored in xylene to clear lipids for 48 h. Finally, tissues were flattened mechanically and suspended in pure methyl salicylate prior to imaging. Due to the large size of whole mammary tissues, overlapping fields of view were captured and stitched together using the "Photomerge" function in Adobe Photoshop. Gaps in the backdrop of the merged images were filled using the "Content aware fill" function in Adobe Photoshop — if and only if the gaps did not overlap any portion of the tissue proper. All original images are available on the Conrad Lab website. To perform statistical comparison of duct morphology between genotypes, measurements of mammary gland ducts were derived from images using AngioTool64 v0.6a[106]. First, a skeleton representation of the branched duct structure is generated from the input image, which is then used to compute a variety of morphological and spatial parameters for branching characterization. Since this software detects the branches by contrast on a black background, the images of whole mount mammary glands of adult mice were transformed into a compatible input using ImageJ 1.51n.

**Hormone measurements.** We collected serum from 9 males (4 wildtype versus 5 knockout; mean age = 103 days) and 16 females in the proestrous stage (7 wildtype versus 9 knockout; mean age = 96 days) via submandibular collection. Each wildtype individual was matched with ≥1 knockout littermate. Female estrous cycle was determined first by external inspection of the vaginal opening, then by vaginal cytology[107]. Vaginal swabs were collected, transferred to a glass slide, and inspected under light microscopy. Proestrus, estrus, metestrus, and diestrus staging was then classified based on the relative counts of leukocytes, nucleated epithelial cells, and cornified epithelial cells. Only confirmed proestrous females were selected for blood draws and hormone measurements. All blood was drawn at approximately the same time of day, clotted for 90 min at room temperature, and centrifuged at $2000 \times g$ for 15 min. Samples were stored at −20 °C prior to hormone measurements. Male samples were quantified for LH/FSH (EMD Millipore) and testosterone (Immuno-Biological Laboratories Inc), and female samples were quantified for LH/FSH and estradiol (CALBIOTECH) by the University of Virginia Ligand Assay and Analysis Core.

**C3 deposition assay.** Testes obtained post-dissection from Csmd1 knockout and wild-type mice were decapsulated and washed in 1× PBS before mincing. Minced tissue was subjected to enzymatic dissociation as described above. The crude cell preparation thus obtained was treated with ACK buffer (Life Technologies) for 5 min at room temperature to lyse erythrocytes present if any in the cell preparation. The isolated cells were incubated in α-C3 (B9) primary antibody (Santa Cruz Biotechnology sc-28294, 1:100) diluted in FACS buffer (1× PBS, 5%FBS, 0.1% Sodium azide) along with 10% Fc block (to minimize non-specific binding and background fluorescence) for 45 min at room temperature, followed by gt α-mouse Alexa488 (Abcam #A11029 1:250) incubation of 90 mins at RT in the dark with 3 washes of ice cold FACS buffer after each antibody incubation. Flow cytometry was performed with an Accuri C6 cytometer (BD Biosciences).

**Reporting summary.** Further information on research design is available in the Nature Research Reporting Summary linked to this article.

## Data availability

Human sample data are from WHI-SHARe (dbGaP Study Accession phs000386.v7.p3), WHISP (dbGaP Study Accession phs000281.v6.p3), UK Biobank (www.ukbiobank.ac.uk), and GEMINI (www.gemini.conradlab.org) and can be obtained upon application. GTEx data were viewed through www.gtexportal.org and ENCODE data were viewed in the UCSC genome browser (genome.ucsc.edu). RNA-seq data generated in this study have been submitted to the Gene Expression Omnibus accession number (GSE136769). We further provide processed RNA-seq RPKM information for all samples as

Supplementary Data. A reporting summary for this Article is available as a Supplementary Information file. The source data underlying Figs. 1–5, 7 and Supplementary Figs. 1, 3, 4, 6, and 9 are provided as a Source Data file. Additional information is available upon request.

## Code availability

Scripts used to perform data analysis are available upon request.

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

## Acknowledgements

We thank all patients and study participants. D.F.C is supported by the National Institutes of Health (R01HD078641 and R01MH101810). A.S.L. was supported by a Distinguished Scholar Award from Washington University School of Medicine. A.C.L. was supported by a fellowship from the Lalor Foundation. We thank Heather Lawson for training and consultation on animal husbandry. We thank Brianne Tabers for technical assistance with animal husbandry. We thank Anne O'Connor and Jo Merriner for performing histology on testis samples, and helpful discussions. We thank Kelle Moley and members of her laboratory (Praba Esakky, Michaela Reid, and Jessica Saben) for consultations on tissue preparations. We thank Tim Schedl, Matt Baum, and Nicole Rockweiler for helpful discussions. We thank Nicholas Ho for technical assistance with image quantification. We thank Seungeun Lee for discussion of SKAT results. We thank Bill Eades and Chris Holley at the Washington University Siteman Flow Cytometry Core (NCI P30 CA91842) FACS services. We thank the Washington University Rheumatic Disease Core (NIHP30AR48335) for providing backcrossed mouse genotypes. We thank the University of Virginia Ligand Assay and Analysis Core (U54 DH28934) for hormone measurements. The WHI program is funded by the National Heart, Lung, and Blood Institute, National Institutes of Health, U.S. Department of Health and Human Services through contracts HHSN268201100046C, HHSN268201100001C, HHSN268201100002C, HHSN268201100003C, HHSN268201100004C, and HHSN271201100004C. This paper was not prepared in collaboration with investigators of the WHI, has not been reviewed and/or approved by the Women's Health Initiative (WHI), and does not necessarily reflect the opinions of the WHI investigators or the NHLBI. Funding for WHI SHARe genotyping was provided by NHLBI Contract N02-HL-64278. The Women's Health Initiative Sequencing Project (WHISP) was funded by Grant Number RC2 HL102924. This study used data from the NHLBI Grand Opportunity Exome Sequencing Project (GOESP). Funding for GO-ESP was provided by NHLBI grants RC2 HL103010 (HeartGO), RC2 HL102923 (LungGO), and RC2 HL102924 (WHISP). The exome sequencing was performed through NHLBI grants RC2 HL102925 (BroadGO) and RC2 HL102926 (SeattleGO).

## Author contributions

D.F.C. devised the study. A.S.L. and D.F.C. led the experimental design. A.S.L. and D.F.C. led the data analysis. A.S.L., A.C.L., M.L., R.M., N.H., K.A.V., R.E.W., J.P.A., M.K.O., and D.F.C. performed data analyses. K.I.A. and D.T.C. contributed to the azoospermia replication study and experimental validation of CNV calls. A.S.L., J.R., A.U., A.C.L., X.W., and R.A.H. performed experiments. A.S.L. and D.F.C. wrote the paper. All authors read and approved the paper.

## Competing interests

The authors declare no competing interests.
