## [Peer Review File · Nature Communications]

Reviewers' Comments:

Reviewer #1:

Remarks to the Author:

The manuscript by Lee et al. describes in great detail the detection of mutations in the CSMD1 gene in male and female infertility, the distribution of de novo mutations around the CSMD1 locus, CSMD1 expression and the phenotype of CSMD1 knockout mice. Overall, it is put together well and tells a convincing story related to both male and female infertility. I do, however, have a number of questions I would like the authors to answer, as well as suggestions to change the content based on the possible use of published work. The breadth of experiments is impressive, sometimes too data rich and difficult to see all the details, but that also makes it difficult to find reviewers that can comment on the entire paper. As a human geneticist, I will comment only on the genetics aspects, and leave the mouse work for others.

First of all the authors describe deletion CNVs in primary ovarian insufficiency as well as azoospermia. While very convincing and statistically sound as shown in Figure 1A, the figure 1B does not make very clear what CNVs were used. It contains much too much information that appears less relevant such as the duplication CNVs (not discussed in the paper?) but also the rare SNV locations which are interesting but probably fit better in separate figure to link with the data presented in figure 1D. This would be useful as the location of these CNVs in both cases and controls is important to illustrate in order to try and understand their mechanisms of action as they affect mostly intronic sequence. Furthermore, a better illustration may explain what is meant with recurrent CNV as figure 1B seems to indicate deletion CNVs of different sizes. Related to this, it is unclear if these CNVs are (maternally?) inherited or occur de novo. Any information on this would be very important given that the deletion CNV point to a loss of function. This is also important to know for the rare SNVs.

In the next paragraph the authors discuss the de novo mutation rare across CSMD1. This part in the paper appears to represent work that is largely/completely published (see citation 54, with Figure 2 displaying CSMD1, with co-authors among the authors). If representing novel work, it would be important to explain what is novel and what was published already in the results section. In addition, there is much more detailed follow-up work published on the enrichment of de novo mutations on chromosome 8 by Johnsson et al in Nature 2017, showing that the hotspot of de novo mutations extends well beyond CSMD1 (in fact is a stretch of 20 Mb). This work is not cited by the authors and that is worrisome. Overall, I miss discussion on this work in the discussion section of the paper, not a single line of discussion is related to this work. Given the overlap with published work, the density of the paper and the fact that this result is not even discussed by the authors, it may better to delete this entire part of the paper.

The rest of the paper seems very well done but I do not have the expertise to judge the mouse work as well as the complement studies. However, I do miss information and discussion on the way CSMD1 variants could cause infertility. From the human work it seems clear the mechanism of action is a loss of function. How does this match with the mouse work? Most work seems to be about the mouse KO models, but are these really relevant to the human findings? The paper would really benefit from a good discussion on this relation between the human and the mouse findings.

Reviewer #2:

Remarks to the Author:

Research summary:

In the manuscript submitted by Lee and colleagues, they performed a series of rare variant analyses across women and men to find genes that contribute to primary gonadal dysfunction. As a result they identified CSMD1 gene that encode a complement regulatory protein as a strong candidate in both sexes. In order to interpret and support their genetic findings, the authors generated Csm1 knockout mice and investigated the reproduction-related phenotypes in both sexes of these mice. Because they have observed significantly increased deposition of complement C3 protein in the testis tissues of Csm1 knockout mice, they further generated Csm1 and C3

double knockout mice, and reported that the double KO leads to a non-additive reduction in fertility. Based on these studies, the authors concluded that that CSMD1 and the complement pathway play an important role in the normal postnatal development of the gonads in both sexes. Comments:

- 1) Overall, the phenotypes of *Csmd1* knockout mice do not support the authors genetic findings in human. Throughout the manuscript, the authors did not present the results of fertility tests (accumulated numbers of pups being born by a KO mother or a KO father mated with WT, in at least six months). The females had only minor defects in terms of ovarian functions, if there are any. As shown in Table 1, the litter sizes were not affected in neither male nor female KO mice.
- 2) Some key analyses of ovarian and testis functions are missing. For example, did the KO females ovulate normal numbers of mature oocytes after superovulation treatment? Did normal numbers of sperms present in the epididymis of the KO males? The histology pictures of testis sections in Figure 5 and Supplementary Figure 6 are at poor qualities.
- 3) The studies throughout the manuscript were unfocused. The authors analyzed the functions of CSMD1 and C3 in testis, ovary, and mammary gland, but the results in all these organs are descriptive, preliminary, and not convincing.
- 4) For most of the parts in this manuscript, the results are descriptive. No molecular functions of CSMD1 and C3 in male or female reproductions were investigated.
- 5) The observed reproduction abnormalities in the double knockout females might be caused by disturbed hemostasis at the whole animal level, instead of primary functional defects in the ovary.
- 6) For detecting localization of CSMD1 in the ovary and testis, immunofluorescence should be performed on tissue sections made from WT and *Csmd1* KO animals side by side to rule out nonspecific signals.

Reviewer #3:

Remarks to the Author:

This study identified rare deletions in the complement regulatory gene CSMD1 associated with male and female infertility. A subset of the deletions were validated using TaqMan assays. Results were replicated in the UK biobank and supported from GWAS studies. RNA sequencing, in situ hybridization and knockout studies were also performed to provide functional support for the role of CSMD1. CSMD1 is enriched at the germ-cell/somatic-cell interface in both male and female gonads. There is evidence for impaired ovarian function and breeding success in knockout females, as well as impaired branching of the mammary glands. There is show severe histological degeneration and significantly increased deposition of complement C3 protein in the testes of knockout males. Double knockout of *Csmd1* and C3 leads to a non-additive reduction in breeding success, suggesting that CSMD1 and the complement pathway play an important role in the normal postnatal development of the gonads in both sexes.

The paper presents provides a comprehensive evaluation and novel findings on the role of CSMD1 in male and female fertility. There is strong evidence for the association of rare deletions with fertility supported by the knockout experiments suggesting absence of CSMD1 has a negative impact on fertility. The manuscript would be improved from some reorganization as there are inconsistencies in emphasis between the title, introduction and discussion. The title and introduction focus on infertility, but much of the discussion is focused on other aspects of the results (mammary gland, brain) and infertility is barely mentioned in the conclusions. The results discussed are important, but implications of the study for fertility also warrant further discussion.

The experiments tell us about negative effects of CSMD1 knockouts on fertility, but less about the role of CSMD1 in fertility regulation when CSMD1 is expressed normally. Can the authors discuss the possible role of intact CSMD1 in regulating fertility? There is evidence for association of three SNPs in CSMD1 with age at menarche (AAM) from GWAS studies. What are the effect sizes for AAM for these common variants? Is it known if there are eQTLs for CSMD1 for these critical SNPs or whether these SNPs affect protein function?

The authors make a strong point about intense recent selection at the CSMD1 locus in humans, but there is no further discussion of the implications of this selection later in the paper. What does the finding of deletions associated with fertility in a region of high selection mean? Are the deletions associated with other phenotype in the UK Biobank data or other studies? What are the reasons for high rates of selection in this region, if not on fertility?

The finding of delayed time to get pregnant, but no difference in litter size is curious? What is the explanation for normal litter size in face of reduced fertility?

Reviewers' comments:

Reviewer #1 (Remarks to the Author):

The manuscript by Lee et al. describes in great detail the detection of mutations in the CSMD1 gene in male and female infertility, the distribution of de novo mutations around the CSMD1 locus, CSMD1 expression and the phenotype of CSMD1 knockout mice. Overall, it is put together well and tells a convincing story related to both male and female infertility. I do, however, have a number of questions I would like the authors to answer, as well as suggestions to change the content based on the possible use of published work. The breadth of experiments is impressive, sometimes too data rich and difficult to see all the details, but that also makes it difficult to find reviewers that can comment on the entire paper. As a human geneticist, I will comment only on the genetics aspects, and leave the mouse work for others.

We appreciate the reviewer's positive reading of the manuscript. We are excited by the multidisciplinary scale and implications of our work, but agree that this makes holistic and rigorous data generation, interpretation, and peer review of the work a challenge. This challenge is reflected in the original structure of the manuscript. In this revision we have attempted to simplify the manuscript by eliminating some sections (as suggested by the reviewer) and thorough restructuring. We have made the narrative more linear, by eliminating the stand-alone sections on the complement-related experiments at the end, and instead, introducing the complement hypothesis earlier in study and evaluating complement activity within each tissue-specific section (testis, ovary, mammary).

First of all the authors describe deletion CNVs in primary ovarian insufficiency as well as azoospermia. While very convincing and statistically sound as shown in Figure 1A, the figure 1B does not make very clear what CNVs were used. It contains much too much information that appears less relevant such as the duplication CNVs (not discussed in the paper?) but also the rare SNV locations which are interesting but probably fit better in separate figure to link with the data presented in figure 1D.

We have reorganized and enhanced this figure to address these points. We switched panels A and B, introducing the map of genetic variation first. We now only show the location of genetic variation in 1A that was used in the test results reported in 1B (i.e. we removed the duplications). We reordered and relabeled the variation tracks and epidemiological results so that they are now in the same order, which we hope better emphasizes the relationships between 1A and 1B. Although we keep the rare SNVs in new Figure 1A, we have replotted them as a stacked dotplot for ease of visualization and to better compare the mutational landscape of SNVs vs. CNVs along the gene.

This would be useful as the location of these CNVs in both cases and controls is important to illustrate in order to try and understand their mechanisms of action as they affect mostly intronic sequence. Furthermore, a better illustration may explain what is meant with recurrent CNV as figure 1B seems to indicate deletion CNVs of different sizes.

The reviewer correctly notes that the association is driven by multiple rare deletions with distinct breakpoints. To avoid confusion, we have removed the word “recurrent” as this does have a special meaning for some in the cytogenetics field, that the same (approximate) breakpoints are observed recurrently across cases, which is not the sense that we intended here.

Please also see our comments to Reviewer 3 regarding possible effects of mutations in intronic sequence.

Related to this, it is unclear if these CNVs are (maternally?) inherited or occur *de novo*. Any information on this would be very important given that the deletion CNV point to a loss of function. This is also important to know for the rare SNVs.

The research subjects in our cohorts were not ascertained as part of families, but instead were unrelated cases and controls. Based on the different deletion breakpoints, we can infer that these deletions are distinct rare variants and not shared polymorphisms. However, without family level data we cannot further discriminate which deletions are *de novo* and which inherited, nor the route of inheritance where that is the case.

Similarly, we cannot definitively confirm *de novo* status for the SNVs. However, motivated by the reviewer’s comment, we tabulated the gnomAD allele frequency of all SNVs in our study. The mean minor allele frequency across all populations is 0.2%, consistent with rare but not necessarily *de novo* mutations.

In the next paragraph the authors discuss the *de novo* mutation rare across CSMD1. This part in the paper appears to represent work that is largely/completely published (see citation 54, with Figure 2 displaying CSMD1, with co-authors among the authors). If representing novel work, it would be important to explain what is novel and what was published already in the results section. In addition, there is much more detailed follow-up work published on the enrichment of *de novo* mutations on chromosome 8 by Jonsson et al in Nature 2017, showing that the hotspot of *de novo* mutations extends well beyond CSMD1 (in fact is a stretch of 20 Mb). This work is not cited by the authors and that is worrisome. Overall, I miss discussion on this work in the discussion section of the paper, not a single line of discussion is related to this work. Given the overlap with published work, the density of the paper and the fact that this result is not even discussed by the authors, it may better to delete this entire part of the paper.

The reviewer’s assessment of the relevant literature is thoughtful and correct. We thank the reviewer and have deleted this entire part of the paper (and related items in the Methods, etc.). The analyses in this section were originally performed years ago and are since then out of date.

To complete the thought here, we hypothesize that the enrichment of *de novo* mutations is generally localized to CSMD1 and that this increase in mutation rate is modulated by *cis* modifiers which corroborate the evolutionary patterns described by Jonsson *et al*. We are

fascinated by the finding that mutation rate over *CSMD1* interacts with parent of origin and maternal age—and that such factors may have a multiplicative effect on fertility. However, the data we have in hand is not sufficient to discuss this further and we removed this section altogether per the reviewer's suggestion.

Please see our comments to Reviewer 3 for additional discussion on selection and potential pleiotropy at this locus.

The rest of the paper seems very well done but I do not have the expertise to judge the mouse work as well as the complement studies. However, I do miss information and discussion on the way *CSMD1* variants could cause infertility. From the human work it seems clear the mechanism of action is a loss of function. How does this match with the mouse work? Most work seems to be about the mouse KO models, but are these really relevant to the human findings? The paper would really benefit from a good discussion on this relation between the human and the mouse findings.

Reviewer 3 had similar thoughts. Please also refer to our comments to reviewer 3 for additional discussion.

In an effort to shorten an already long manuscript we initially omitted a more detailed discussion on the effect of *CSMD1* on fertility in the first submission. We too favor a loss of function mode of action, which we now discuss in detail in the context of functional constraint. We have now added several paragraphs to the discussion relating the human and mouse findings. We have also heavily restructured the paper to better integrate the mouse and human work, starting with the introduction, where we clarify our hypothesis that *CSMD1* mutation is mediating gonad function through regulation of the complement system. More explicitly, we favor a scenario in which loss of *CSMD1* function leads to overzealous complement-mediated phagocytosis in the testes, ovary, and mammary—a disease process paralleled in schizophrenia through a process known as synaptic pruning.

In this system, capturing direct, *bona fide* evidence of excessive germ cell phagocytosis in the *Csmd1* knockout is exquisitely difficult, especially because it is a transient process. This difficulty is compounded by the fact that the timing of the *Csmd1* KO gonad degeneration is variable from mouse to mouse. As a point of comparison, normal synaptic pruning in the mouse retinogeniculate system occurs during a highly stereotyped and predictable time window (postnatal day 5 exactly), making it far more tractable for experimental sampling (see Shaefer *et al.*, <https://doi.org/10.1016/j.neuron.2012.03.026>).

Reviewer #2 (Remarks to the Author):

Research summary:

In the manuscript submitted by Lee and colleagues, they performed a series of rare variant analyses across women and men to find genes that contribute to primary gonadal dysfunction. As a result they identified *CSMD1* gene that encode a complement regulatory protein as a strong candidate in both sexes. In order to interpret and support their genetic findings, the authors generated *Csmd1* knockout mice and investigated the reproduction-

related phenotypes in both sexes of these mice. Because they have observed significantly increased deposition of complement C3 protein in the testis tissues of *Csmd1* knockout mice, they further generated *Csmd1* and C3 double knockout mice, and reported that the double KO leads to a non-additive reduction in fertility. Based on these studies, the authors concluded that that *CSMD1* and the complement pathway play an important role in the normal postnatal development of the gonads in both sexes.

Comments:

1) Overall, the phenotypes of *Csmd1* knockout mice do not support the authors genetic findings in human.

We acknowledge the reviewer's legitimate criticism that our phenotypic characterization of the *Csmd1* KO is largely, but not solely, descriptive. Furthermore, we do not assert that the mouse phenotypic testing would stand alone to robustly implicate *CSMD1* in human infertility or warrant publication in Nature Communications. However, this interpretation ignores the larger context of the novel, unbiased human genetic findings, which implicated *CSMD1* without any prior assumptions about the biology. In this context, the mouse data confirms the human data and takes significant steps towards defining the mechanism of pathogenesis.

Furthermore, we politely disagree that the *Csmd1* KO phenotypes do not support its role in gonadal function. Although some of the phenotypes are either subtle and/or heterogeneous (e.g., histological degeneration, time to pregnancy), they are statistically and reproducibly distinct between wildtype and knockout tissues and cannot be explained by normal phenotypic variability.

We also reiterate the polygenic, complex genetic architecture of common infertility in humans. Here we report an odds ratio of 16 for in our female patient cohort, indicating just a 16-fold increased risk of early idiopathic menopause for individuals bearing a *CSMD1* deletion (as a point of comparison, a classic monogenic Mendelian mutation, even with reduced penetrance, would have an odds ratio in the hundreds). By our own calculation, we expect instances of *CSMD1* mutations that do not lead to disease, and even still more instances of disease individuals who do not bear a *CSMD1* mutation.

When taken in the context of the ascertained human phenotype, we do not expect *Csmd1* KO mice to be sterile or necessarily even have reduced litter sizes when young. Our human patients with the *CSMD1* deletion could complete full child-bearing and subsequently undergo early idiopathic menopause prior to age 40. Rather, as supported by the mouse data, *Csmd1* loss-of-function is a clinically relevant risk factor for reduced reproductive lifespan, and thus, infertility.

We appreciated the reviewer's valuable insights on appropriate fertility analyses and incorporated as many suggestions as we could with the resources available (documented in comments below).

Throughout the manuscript, the authors did not present the results of fertility tests (accumulated numbers of pups being born by a KO mother or a KO father mated with WT, in at least six months).

We apologize for this omission and have included the results of single KO fertility tests as requested in Fig. 3 and Fig. 4, in addition to the double KO fertility data already presented.

The females had only minor defects in terms of ovarian functions, if there are any. As shown in Table 1, the litter sizes were not affected in neither male nor female KO mice.

We agree that the ovarian defects are subtle—specifically the increased time to pregnancy. They do differ significantly between wildtype and knockout females. This result is however, consistent with the human definition of premature ovarian failure i.e. shortened reproductive lifespan, rather than sterility. and are not necessarily inconsistent with the infertility phenotypes ascertained in our human genetic studies. Ultimately we Please see comments above.

2) Some key analyses of ovarian and testis functions are missing. For example, did the KO females ovulate normal numbers of mature oocytes after superovulation treatment? Did normal numbers of sperms present in the epididymis of the KO males?

We appreciate the reviewer's suggestion. In our revised manuscript we have measured daily sperm production, an assay we have previously performed to quantify testis function in other transgenic mouse models (see Cotton *et al.*, <https://doi.org/10.1242/jcs.02704>). We now show that there is a 25% reduction in daily sperm production in KO males compared to WT (see Figure 3B).

We did not perform superovulation experiments, although we agree such data would strengthen the phenotyping. Instead, we did perform histological follicle counting on 95 animals, summarized in Figure 4D and Methods, which shows a significant difference in proportion of atretic and pre-ovulatory follicles. We believe these data at least partially provides the information from superovulation.

The histology pictures of testis sections in Figure 5 and Supplementary Figure 6 are at poor qualities.

We are appreciative of the reviewer's attention to quality. As requested we have improved these pictures. We have generated new histology images for Figure 5 (now Figure 3). We started over from the beginning: we have cut new sections and performed new staining using an improved protocol which we have used previously with good results (see Dunleavy *et al.*, <https://www.ncbi.nlm.nih.gov/pubmed/29136647>). For Supplementary figure 6 (now 5) we have replaced the original histology images with higher resolution versions.

3) The studies throughout the manuscript were unfocused. The authors analyzed the functions of CSMD1 and C3 in testis, ovary, and mammary gland, but the results in all these organs are descriptive, preliminary, and not convincing.

4) For most of the parts in this manuscript, the results are descriptive. No molecular functions of CSMD1 and C3 in male or female reproductions were investigated.

We respectfully disagree. Please see response to comment 1 above. Studies that implicate a previously unknown locus to a common human disease and simultaneously put forth mechanistic insights reflect truly rare, seminal advances (see Genovese *et al.*, <https://doi.org/10.1126/science.1193032> and Sekar *et al.*, <https://doi.org/10.1038/nature16549> for examples). Although the reviewer's criticisms are legitimate, we do not consider this to be a reasonable standard in evaluating our findings. Throughout the course of the project, we purposefully prioritized establishing descriptive links across all three tissues—little of which existed previously—over molecular mechanism in a single tissue which we estimated would take years (on top of the years spent producing a convincing human genetic study). Furthermore, the provision of mouse phenotype data is of enormous value clinically, as they form part of the knowledgebase that human geneticists will use to make a diagnosis and interpret CSMD1 variants in the future, and are a critical step towards defining mechanism.

5) The observed reproduction abnormalities in the double knockout females might be caused by disturbed hemostasis at the whole animal level, instead of primary functional defects in the ovary.

The reviewer is raising the concern that double KO of CSMD1 and C3 may lead to secondary female infertility due to problems with hemostasis, instead of primary defects in the ovary. Although possible, we do not favor this scenario for two reasons. First, we find equally strong reproductive defects in the male double KOs; second, we saw no similar reproductive problems in the C3 KO females or males. Nonetheless, the reviewer raises a legitimate point that, if CSMD1 is expressed in multiple cell types, our interpretations should consider the possibility that pathology may involve other tissues. Based on inspection of the GTEx Consortium RNA-seq dataset, which surveys >30 tissues from over 900 post-mortem human donors, it appears that CSMD1 is most strongly expressed in testis and brain (arrows):

We note that the RNA-seq of bulk ovarian tissue, such as reported by GTEx, is underpowered to detect transcripts with expression restricted to rare ovarian cells types like oocytes or cumulus cells. We have added the caveat to our discussion of the double KO that disruption of neuronal cells could also play a role in pathology.

6) For detecting localization of CSMD1 in the ovary and testis, immunofluorescence should be performed on tissue sections made from WT and *Csmd1* KO animals side by side to rule out nonspecific signals.

We have provided a new supplementary figure that provides detailed characterization of molecular consequences of the *Csmd1*/tm1Lex mutation, as well as side-by-side comparison of antibody staining between KO and wildtype. There are no published monoclonal antibodies available for CSMD1. Staining results for our antibody indicate that there is staining in the mutant line; we hypothesize this is likely binding a CSMD1 paralog (CSMD2 or CSMD3, which are highly homologous to CSMD1). Using X-gal or anti-B Gal staining we have shown that the CSMD1 promoter drives expression of LacZ in testicular germ cells and specifically in developing follicles in the ovary (Supplementary Figure 4). This is consistent with our anti-CSMD1 IF. It is also consistent with previous descriptions of *CSMD1* localization using RNA in-situ (see Kraus *et al.*, <https://doi.org/10.4049/jimmunol.176.7.4419>, and image below).

CsmD1* mRNA *in situ* hybridization in mammary and ovary. Adapted from Kraus *et al.

Reviewer #3 (Remarks to the Author):

This study identified rare deletions in the complement regulatory gene CSMD1 associated with male and female infertility. A subset of the deletions were validated using TaqMan assays. Results were replicated in the UK biobank and supported from GWAS studies. RNA sequencing, *in situ* hybridization and knockout studies were also performed to provide functional support for the role of CSMD1. CSMD1 is enriched at the germ-cell/somatic-cell interface in both male and female gonads. There is evidence for impaired ovarian function and breeding success in knockout females, as well as impaired branching of the mammary glands. There is show severe histological degeneration and significantly increased deposition of complement C3 protein in the testes of knockout males. Double knockout of *CsmD1* and C3 leads to a non-additive reduction in breeding success, suggesting that CSMD1 and the complement pathway play an important role in the normal postnatal development of the gonads in both sexes.

The paper presents provides a comprehensive evaluation and novel findings on the role of CSMD1 in male and female fertility. There is strong evidence for the association of rare deletions with fertility supported by the knockout experiments suggesting absence of CSMD1 has a negative impact on fertility. The manuscript would be improved from some reorganization as there are inconsistencies in emphasis between the title, introduction and discussion. The title and introduction focus on infertility, but much of the discussion is focused on other aspects of the results (mammary gland, brain) and infertility is barely

mentioned in the conclusions. The results discussed are important, but implications of the study for fertility also warrant further discussion.

We warmly appreciate the reviewer's endorsement of our genetic findings and importance of our results. We agree with the reviewer's criticisms on the structuring of the paper; we initially kept the discussion concise as we wanted to avoid over interpretation, but we were perhaps too cautious. We have now added implications for the study of infertility. More generally, we have heavily restructured the manuscript to better harmonize the title, introduction, and discussion. This was an extremely useful criticism, as we feel that the restructured manuscript better emphasizes the biological impact of complement regulation, which were at times buried in the complicated text.

Reviewer 1 also had similar concerns. Please refer to comments to Reviewer 1 for additional details.

The experiments tell us about negative effects of *CSMD1* knockouts on fertility, but less about the role of *CSMD1* in fertility regulation when *CSMD1* is expressed normally. Can the authors discuss the possible role of intact *CSMD1* in regulating fertility? There is evidence for association of three SNPs in *CSMD1* with age at menarche (AAM) from GWAS studies. What are the effect sizes for AAM for these common variants? Is it known if there are eQTLs for *CSMD1* for these critical SNPs or whether these SNPs affect protein function?

For further discussion on *CSMD1* in disease, please see comments to Reviewer 1. Briefly, we favor a scenario in which complement is used by the body to tag defective or excess germ cells for phagocytosis in normal reproductive function, and *CSMD1* is a necessary safeguard to control complement activation on self-cells. Loss of *CSMD1* function leads to overzealous complement deposition and subsequent inappropriate phagocytosis of germ cells in both testes and ovary. This is now detailed in the discussion.

We have added some text to the discussion that describes the possible role of intact *CSMD1* on fertility regulation. We have also factored in this discussion publicly available functional genomics data such as eQTL information from GTEx and epigenome data from ENCODE. One important limitation of these public genome projects is that they typically neglect reproductive cell types (e.g. ENCODE), or the reproductive tissues are processed in bulk, which can occlude signals from rare cell types (e.g. GTEx).

As we mention (and visualize) in the response to reviewer #2 above, GTEx indicates that *CSMD1* is most highly expressed in brain and testis. As the reviewer is no doubt aware, GTEx interrogates over 40 tissues (and cell lines) for eQTLs. **There are only two tissues with *CSMD1* eQTLs: testis and thyroid.** Although the gene body spans over 2Mb of the genome, all of these QTLs are clustered within a 390Kb window spanning introns 1 and 2: the same region that contains the majority of our fertility-associated deletions, as well as 2 of the 3 GWAS peaks from the AAM study by Day *et al* :

<https://gtexportal.org/home/bubbleHeatmapPage/ENSG00000183117.13>

The plot above shows the location of CSMD1 eQTLs and their normalized effect sizes (NES). None of the lead SNPs from the Day *et al.* study appear to be in appreciable LD with CSMD1 eQTLs identified by GTEx. All three lead SNPs are intronic and non-coding. Please see <https://gtexportal.org/home/bubbleHeatmapPage/CSMD1> for an interactive demonstration. Here are the Day *et al.* rsIDs for quick reference: rs2688326, rs2724961, rs4875424.

The effect sizes of the common risk alleles from Day *et al.* are very modest as may be expected. The regression coefficients (beta y/allele) range from -0.03 to -0.045 (see Figure 1). As a point of comparison our estimated regression coefficient for our WHI association was 3.0.

To gain further clarity about the possible functional consequences of the disease-associated deletions, we overlaid them on ENCODE functional genome segmentations available for human cell lines in UCSC Human Genome Browser build GRCh37. These segmentations summarize multidimensional functional genome data generated by ENCODE into a single likely functional annotation, e.g. “enhancer”, “heterochromatin”, “promoter” etc. Such tracks were available for 6 human cell lines. While many annotations may be cell-type specific, we noted that there were 4 “consensus” CTCF binding sites that were observed in all or nearly all of the cell lines (circled below). Some of the CSMD1 intronic deletions from cases overlapped these consensus sites, but not all of them:

As we mention, such an analysis really needs to be performed using functional data from the relevant human cell types, ideally at single-cell resolution which don't exist at the moment.

The authors make a strong point about intense recent selection at the CSMD1 locus in humans, but there is no further discussion of the implications of this selection later in the paper. What does the finding of deletions associated with fertility in a region of high selection mean? Are the deletions associated with other phenotype in the UK Biobank data or other studies? What are the reasons for high rates of selection in this region, if not on fertility?

Please also see our comments to reviewer 1, who had similar questions.

Just to clarify, we noted that CSMD1 is located in a region of intense germline mutation across all/most humans ('exceptional evolution'), but we are still agnostic about whether this mutation is attributable to selective forces. Our work, along with previous published works indicate that the accelerated evolutionary tempo of the larger CSMD1 locus is driven by increased SNV mutation rates. This signature is maintained among human, chimp and gorilla, but absent among orangutans (see phylogeny below). Further complicating this increase in mutational input is the finding that CSMD1 is under fierce constraint against protein coding changes (see newly added RVIS and pLI discussion in the manuscript), potentially due to their effect on fertility. These oppositional forces could ultimately generate signatures of pseudo-balancing or purifying selection over CSMD1, where we observe an excess of rare or intermediate variation. Due to the length of the manuscript and overlap with prior work, we cut this section per request of Reviewer 1.

CSMD1 (red dotted lines) nucleotide diversity spikes in human, gorilla, chimp, and bonobo—but not orangutan.

We share the referee’s intuition that *CSMD1*—and by extension complement—is a candidate for pleiotropic effects across several phenotypic dimensions. This is a point of impact which we hope to convey through our restructuring of the manuscript.

Dysregulation of complement is a core feature of several heritable diseases beyond infertility, perhaps most famously in schizophrenia. Motivated by the referee’s comment, we performed a single locus association between rare *CSMD1* CNVs and i) age-related macular degeneration and ii) Alzheimer’s disease within our original human early idiopathic menopause cohort. Extensive prior work has established a relationship between both AMD and AD pathogenesis and complement dysregulation. Remarkably, we identified a significant relationship between rare *CSMD1* deletions and AMD (OR = 3.5; p-value = 0.03)—though not with AD (P = 0.6).

We do not feel this putative association meets the statistical standard to be included in our published manuscript in the absence of large-scale replication cohorts (we assembled our cohorts exclusively to study infertility). Nevertheless, we are tantalized by the possibility that other diseases with known complement-mediated pathology such as atypical HUS, hereditary angioedema, and C3 glomerulonephritis may show similar relationships with *CSMD1* mutations.

The finding of delayed time to get pregnant, but no difference in litter size is curious? What is the explanation for normal litter size in face of reduced fertility?

We agree. One possible explanation is a slower, or erratic, estrous cycle in the KO. Other explanations could entail stochastic factors that act in an all/nothing kind of manner; for instance if there were some phenomenon that resulted in complete failure to ovulate or complete loss of a litter, such as atresia/phagocytosis of all follicles in a cycle, or phagocytosis/absorption of oocytes/zygotes in the fallopian tubes (we did see consistent, severe tubal degeneration in double KOs).

Reviewers' Comments:

Reviewer #1:

Remarks to the Author:

The authors have responded in great detail to my initial review. Most importantly, they have extensively reorganizing the manuscript by deleting parts that were previously published and not contributing in detail to the main conclusions. Also, they have integrated the mouse and human data in a more clear manner. All of this greatly contributes to the readability of this very interesting paper. I have no further remarks.

Reviewer #2:

Remarks to the Author:

Summary:

In this revised manuscript, the authors carefully addressed comments and critics raised by 3 reviewers including me. Their efforts to improve the current study is appreciated. Based on human genetics data, they performed a series of RNA variant analyses to look for previously unknown genes that potentially contribute to decreased fertility in both sexes, and identified CSMD1, which encodes a complement regulatory protein, as a strong candidate. Because the role of CSMD1 in mammalian gonads has never been reported before, the authors investigated the expression, tissue specific localization of CSMD1 in mouse and human gonads. To collect further evidence that CSMD1 is a factor associated with normal fertility, they studied the reproduction-related phenotypes as well as gonad functions in previously described *Csmd1* knockout mice. The results showed that *Csmd1* knockout testes and ovaries reduces in morphologic quality and reproductive performance. These studies provide new knowledge regarding to genetic bases of reduced fertility in human, and suggest the potential involvement of complement-related proteins in regulating mammalian gonad functions.

I agree with the comments of Reviewer 1, "The breadth of experiments is impressive, sometimes too data rich and difficult to see all the details, but that also makes it difficult to find reviewers that can comment on the entire paper." As a biologist of reproduction and development, I will comment mainly on the mouse work, and leave the human genetics aspects for others.

Specific comments:

1: In general, I think the authors addressed by previous comments appropriately. In my previous comments, I indicated that the effect of *Csmd1* knockout on fertility were relatively subtle in female mice. The authors explained in the revised text that "The female phenotype we ascertained in human was not infertility per se, but idiopathic early menopause, which is difficult to model in mice. Thus, we searched for more subtle gonadal defects in the knockout background." I think this explanation is justified by the data. But I still think the superovulation assay should be performed in WT and KO mice, because this is a standard assay for evaluation of mammalian ovarian function. The readout is more direct and clear than the ovarian morphology analyses such as follicle counting. This is an easy experiment to do and should not be time-consuming. Or maybe the results did not show differences between WT and KO group so the authors are reluctant to present?

2: Line 194-196 and Fig. 2C-D: "Throughout oogenesis, CSMD1 shows lower expression in early follicles (i.e., primordial and primary follicles) and higher expression in late follicles (i.e., secondary, tertiary, and pre-ovulatory follicles." Does the author mean CSMD1 protein is expressed in the whole follicle (oocyte, granulosa cells, and theca cells) or only in oocytes and theca cells? I did not see positive signals in granulosa cells. Please specify the cell types. Maybe they should say: "CSMD1 shows lower expression in oocytes of early follicles (i.e., primordial and

primary follicles) and higher expression in oocytes of late follicles..."

In addition, I think the scale bar in the right panel of Fig. 2D is labeled wrong. The corpus luteum cannot be so small.

3: In response to my previous suggestion, the authors provided CSMD1 immunofluorescence results in KO testes and ovaries. As I worried, there are still fluorescent signals on tissue sections made from KO animals. The authors gave explanations for this unexpected results, which is acceptable to me. However, this seemingly discrepancy may also concerns future readers, not just me. Therefore, I suggest the authors to describe the result in Supplementary Figure S4C with more details in the text and include their interpretations of this result in the Discussion section.

Reviewer #3:

Remarks to the Author:

The authors have addressed comment by all reviewers. They have restructured the text and substantially improved the manuscript. The discussion might be improved further by balancing comments on effects on male and female infertility.

Reviewers' comments:

Reviewer #1 (Remarks to the Author):

The authors have responded in great detail to my initial review. Most importantly, they have extensively reorganizing the manuscript by deleting parts that were previously published and not contributing in detail to the main conclusions. Also, they have integrated the mouse and human data in a more clear manner. All of this greatly contributes to the readability of this very interesting paper. I have no further remarks.

We thank Dr. Veltman for his rigorous critiques and positive reading of the manuscript.

Reviewer #2 (Remarks to the Author):

Summary:

In this revised manuscript, the authors carefully addressed comments and criticisms raised by 3 reviewers including me. Their efforts to improve the current study are appreciated. Based on human genetics data, they performed a series of RNA variant analyses to look for previously unknown genes that potentially contribute to decreased fertility in both sexes, and identified CSMD1, which encodes a complement regulatory protein, as a strong candidate. Because the role of CSMD1 in mammalian gonads has never been reported before, the authors investigated the expression, tissue-specific localization of CSMD1 in mouse and human gonads. To collect further evidence that CSMD1 is a factor associated with normal fertility, they studied the reproduction-related phenotypes as well as gonad functions in previously described *Csmd1* knockout mice. The results showed that *Csmd1* knockout testes and ovaries reduce in morphologic quality and reproductive performance. These studies provide new knowledge regarding the genetic bases of reduced fertility in human, and suggest the potential involvement of complement-related proteins in regulating mammalian gonad functions.

We thank the reviewer for their rigorous, detail-oriented critiques and for acknowledging our efforts to incorporate these critiques to improve our study.

I agree with the comments of Reviewer 1, "The breadth of experiments is impressive, sometimes too data rich and difficult to see all the details, but that also makes it difficult to find reviewers that can comment on the entire paper." As a biologist of reproduction and development, I will comment mainly on the mouse work, and leave the human genetics aspects for others.

Specific comments:

1: In general, I think the authors addressed my previous comments appropriately. In my previous comments, I indicated that the effect of *Csmd1* knockout on fertility was relatively subtle in female mice. The authors explained in the revised text that "The female phenotype we ascertained in human was not infertility per se, but idiopathic early menopause, which is difficult to model in mice. Thus, we searched for more subtle gonadal defects in the knockout background." I think this explanation is justified by the data. But I still think the superovulation assay should be performed in WT and KO mice, because this is a standard assay for evaluation of mammalian ovarian function. The readout is more direct and clear than the ovarian morphology analyses such as follicle counting. This is an easy experiment to do and should not be time-consuming. Or maybe the results did not show differences between WT and KO group so the authors are reluctant to present?

We did not attempt the superovulation assay, although the reviewer is correct that it is a common readout for infertility. In retrospect, this was an oversight at the time of the initial study, as we were simultaneously prioritizing i) male histology + breeding performance; ii) female histology + breeding performance; and iii) mammary histology

+ pup survival in parallel. Our group has since changed physical locations and during this transition we are not actively maintaining the mouse line to allow for the superovulation assay in a timely fashion. We apologise for not addressing the reviewer's critique squarely.

Alternatively, after consultation with the editor, we have instead performed more detailed phenotypic analyses of the human WHI and UKB female cohorts as a more subtle quantitative reproductive trait than simply cases vs. controls.

In a well-powered multivariate analysis (n= approx. 76,000 subjects with complete data), we now show that *CSMD1* 5'-deletions reveal a statistically significant association with age at last birth ($p = 4.1 \times 10^{-3}$). The model effect size is substantial, with the risk allele estimated to decrease the age at last birth by approx. 1 year (new Figure 1E). Using the same model, we estimate this is approximately the same effect size of being a smoker compared to a "never smoker" (1 year reduction) and a larger effect size than being obese (0.6 years reduction). We have also tabulated finer scale human phenotypes for available female cases bearing the *CSMD1* 5'-deletions to better intuit phenotype-genotype correlations (new Table S2). All of the mutations are likely partial loss-of-function. As with the mouse models, each of these index cases had completed pregnancies prior to early gonadal failure.

We believe this analysis will draw additional parallels to consolidate the subtlety of the mouse and human phenotypes and resolve some seemingly paradoxical findings (e.g., Why do pre-sterility *CSMD1* KO mothers give birth to normal numbers of offspring per pregnancy?).

2: Line 194-196 and Fig. 2C-D: "Throughout oogenesis, *CSMD1* shows lower expression in early follicles (i.e., primordial and primary follicles) and higher expression in late follicles (i.e., secondary, tertiary, and pre-ovulatory follicles." Does the author mean *CSMD1* protein is expressed in the whole follicle (oocyte, granulosa cells, and theca cells) or only in oocytes and theca cells? I did not see positive signals in granulosa cells. Please specify the cell types. Maybe they should say: "*CSMD1* shows lower expression in oocytes of early follicles (i.e., primordial and primary follicles) and higher expression in oocytes of late follicles..."
In addition, I think the scale bar in the right panel of Fig. 2D is labeled wrong. The corpus luteum cannot be so small.

We appreciate and agree with the reviewer's attention to detail here. We have changed the text to be more precise about the cell types of interest in the latest version of the manuscript, now at lines 188-193 and the Figure 2C Legend.

The reviewer's observation on the scale bar in Panel 2D is detailed and correct. We have entirely re-imaged a corpus luteum with the correct scale bar (See new Panel 2D). We also include multiple follicles in the same field of view in order to allow the reader to more easily judge the relative sizes and *CSMD1* expression levels among the corpus luteum and follicles. We thank the reviewer for this comment which improves the quality of our figure.

3: In response to my previous suggestion, the authors provided *CSMD1* immunofluorescence results in KO testes and ovaries. As I worried, there are still fluorescent signals on tissue sections made from KO animals. The authors gave explanations for this unexpected results, which is acceptable to me. However, this seemingly discrepancy may also concerns future readers, not just me. Therefore, I suggest the authors to describe the result in Supplementary Figure S4C with more details in the text and include their interpretations of this result in the Discussion section.

We acknowledge the reviewer's appropriate attention to this detail and do not wish to understate this complicating but important detail. To address the reviewer's comment, we have added more detailed description of the KO immunofluorescence results in the

Results (lines 233-234) as well as our interpretation of the KO signal in the context of loss-of-function intolerance in the Discussion (lines 445-453).

Briefly here, we find that the wildtype patterns of CSMD1 expression are concordant across multiple modalities (e.g., bulk RNA seq, immunofluorescence, and B-Gal reporter expression). In *Csmd1* knockouts, we note residual expression of *Csmd1* mRNA in KO testes, but not KO ovaries. Finally, we note nonspecific antibody signal of CSMD1 protein in KO testes, KO ovaries, and KO mammary. Taking into account the exceptional loss-of-function constraint over *CSMD1* in humans, we believe that the immunofluorescence signal in KO tissues represents paralogous cross-reactivity (e.g., CSMD2/CSMD3) and/or residual native CSMD1 expression due to incomplete mouse *Csmd1* loss-of-function. This result complicates but is consistent with our interpretation of the subtle mouse KO phenotypes.

Reviewer #3 (Remarks to the Author):

The authors have addressed comment by all reviewers. They have restructured the text and substantially improved the manuscript. The discussion might be improved further by balancing comments on effects on male and female infertility.

We thank the reviewer for their rigorous critiques and positive reading of the manuscript. Simply due to the length of the manuscript, we will not add further text to the Discussion.

Reviewers' Comments:

Reviewer #1:

Remarks to the Author:

The additional analysis looking at the association between CSMD1 deletions and age of birth last child in the UK biobank data further confirms the link between CSMD1 deletions and reduced reproductive lifespan.

Reviewer #2:

Remarks to the Author:

The authors have addressed satisfactorily the issues raised in the initial review. This manuscript is to be recommended for publication.

REVIEWERS' COMMENTS:

Reviewer #1 (Remarks to the Author):

The additional analysis looking at the association between CSMD1 deletions and age of birth last child in the UK biobank data further confirms the link between CSMD1 deletions and reduced reproductive lifespan.

We kindly thank Dr. Veltman.

Reviewer #2 (Remarks to the Author):

The authors have addressed satisfactorily the issues raised in the initial review. This manuscript is to be recommended for publication.

We kindly thank Reviewer #2.